# FEATURE MAP MATTERS IN OUT-OF-DISTRIBUTION DETECTION

## ABSTRACT

Detecting and rejecting *out-of-distribution* (OOD) data can improve the reliability and reduce potential risks of a model (e.g., a neural network) during the deployment phase. Recent post-hoc OOD detection methods usually focus on analyzing hidden features or prediction logits of the model. However, feature maps of the backbone would also contain important clues for discriminating the OOD data. In this paper, we propose an OOD score function *Feature Sim* (FS) that can efficiently identify the OOD data by only looking at the feature maps. Furthermore, a novel *Threshold Activation* (TA) module is proposed to suppress non-critical information in the feature maps and broaden the divergences between foreground and background contexts. We provide a theoretical analysis to help understand our methods. The experimental results show that our methods FS+TA and FS+TA+ASH can achieve state-of-the-art on various benchmarks. More importantly, since our method is based on feature maps instead of hidden features or logits, it can be easily adapted to more scenarios, such as semantic segmentation and object detection. The codes are available at Appendix F.

## 1 INTRODUCTION

Modern machine learning models are often trained under a closed-world assumption (Krizhevsky et al., 2017), i.e., the model inputs in the training and testing stages are independently and identically distributed. These inputs are also referred to as *in-distribution* (ID) data. When the models are deployed to open-world scenarios, they will encounter unknown classes (Hendrycks & Gimpel, 2017), which are often referred to as *out-of-distribution* (OOD) data that the models will become overconfident in. This model-overconfidence phenomenon reveals that these models might be harmful to practical applications. To avoid such problems, detecting and rejecting OOD data is critical, which has received more and more attention from researchers (Hsu et al., 2020; Wang et al., 2021b; 2022b; Huang et al., 2021; Sun et al., 2021; Wang et al., 2021a).

Recently, most post-hoc OOD detectors develop their algorithms by analyzing the interaction between network classifiers and data. Specifically, Hendrycks & Gimpel (2017); Liang et al. (2018); Liu et al. (2020); Huang et al. (2021) measure the data uncertainties based on the predicted probabilities or output logits. Sun et al. (2021); Sun & Li (2022) further explore the hidden features and parameters of the classifier's penultimate layer to figure out why the network may be overconfident in OOD data.

However, the methods above only explore features in the classification stage of convolutional neural networks. These features would pass through the *global average pooling* (GAP) layer and lose a lot of spatial information, which also carries clues to detect OOD data (see Fig. 1). In particular, we find that the difference between foreground and background activations in the feature maps can be used to measure the OOD-ness of the data. The foreground indicates the part of the image that is semantically identical to its label, while the background refers to the part of the image that is semantically unrelated to its label. As shown in Figs. 1a and 1c, ID data has more significant foreground activations than OOD data. The reason is that the network is trained based on the features and label space of ID data, so the network would be more familiar with ID data and generate more salient responses.

In this paper, we focus on the difference between spatial activation intensities of ID and OOD data (called *activation difference* in the following paper) and design a score function *Feature Sim* (FS) to accomplish OOD detection without relying on the classifier's features (see Eq. (3)). We use the mean absolute deviation of feature maps to measure features' self-similarity and discriminate OOD

| (a) ID heatmap w/o TA | (b) ID heatmap w/ TA | (c) OOD heatmap w/o TA | (d) OOD heatmap w/ TA |

Figure 1: Feature visualization of ResNet50. The red part of the image represents higher feature activation. Threshold Activation (TA) is our proposed module to suppress features in the middle layer. Compared vertically, ID features have higher activation than OOD features. Compared horizontally, ID features retain more significant activation than OOD features after TA. Therefore, feature maps contain important clues for OOD detection. More visualizations can be found in Appendix C.1.

data, i.e., the absolute difference between the feature map and its mean value. As mentioned above, the activation difference between the foreground and background of OOD data is small, so the mean absolute deviation will also be small, and vice versa. In this way, the network can achieve OOD detection by only looking at feature maps.

The previous works (Goodfellow et al., 2015; Liang et al., 2018) imply that ID features are more robust than OOD features when the input data are perturbed. We further investigate this phenomenon when considering feature maps in OOD detection. As shown in Figs. 1b and 1d, when we suppress the features in the middle layer, there is a significant decrease in OOD features while ID features are still relatively stable. Based on this phenomenon, we propose the *Threshold Activation* (TA) module (see Eq. (5)), which suppresses the features in the middle layer so that the activations of OOD features are greatly weakened without affecting the representation ability of ID features. The Threshold Activation module can amplify the difference between ID and OOD data and improve the OOD detection performance of Feature Sim.

We provide a detailed theoretical analysis to help understand our method FS. We also evaluate our method FS+TA on a series of OOD detection benchmarks. In addition, we compound our method FS+TA with previous methods to fuse the information from the feature map space and the classifier output space to detect OOD data. Experiment results show that the composite method FS+TA+ASH can significantly outperform the separate method and achieve state-of-the-art (95.97% AUROC and 19.56% FPR95) on ImageNet benchmark (Huang et al., 2021). More importantly, our method FS+TA can be adapted to more tasks whose networks are not always equipped with image-level classification heads, such as object detection and semantic segmentation (see Table 6). Therefore, more models may benefit from our method to gain the ability to detect OOD inputs without purposely designed OOD detectors. In summary, our key contributions are as follows:

- We propose a simple yet effective method Feature Sim (see Eq. (3)), to accomplish post hoc OOD detection by only looking at the feature maps of the backbone.
- We propose a plug-in module Threshold Activation (see Eq. (5)) to improve the performance of OOD detection. It pulls apart the divergence between ID and OOD data by suppressing features in the middle layer, significantly improving Feature Sim.
- Our method can collaborate well with previous methods and fuse information from both the feature maps and classifier. Experimental results show that FS+TA+ASH can achieve state-of-the-art on various OOD detection benchmarks.

## 2 RELATED WORK

In this section, we focus on post-hoc OOD detection (Liu et al., 2020; Lee et al., 2018; Sun et al., 2021; Wang et al., 2021a), which do not change the model's parameters in the inference time.

**Output-based methods** depend on the predictions or logits output by the classifier. MSP (Hendrycks & Gimpel, 2017), Maxlogit (Hendrycks et al., 2022) and MCM (Ming et al., 2022a) propose to use the maximum probability or logit of model prediction to detect OOD data. ODIN (Liang et al., 2018) adds temperature scaling to the softmax function and perturbs the inputs to improve the separability of ID and OOD data. Different from the previous methods, Energy (Liu et al., 2020) uses the energy function (LeCun et al., 2006) instead of the softmax function to accomplish OOD detection.

**Feature-based methods** rely on features of the network and explore latent useful representations for OOD detection. ReAct (Sun et al., 2021) clips noisy activations of the classifier's penultimate layer

Figure 2: The pipeline of our method. The orange parts highlight the proposed Threshold Activation and Feature Sim operations. An input image first goes through the shallow layers of a convolutional neural network. Then the middle layer features are processed by Threshold Activation (TA). The processed features continue the forward propagation through the deep layers. Finally, we calculate the mean absolute deviation of the output feature map as the Feature Sim score. The input image shown above belongs to ID classes, so after TA, the activation is still concentrated on the foreground and results in a higher Feature Sim score.

to improve OOD detection performance. Differently, DICE (Sun & Li, 2022) choose to selectively use the most salient weights of the classifier's penultimate layer for OOD detection. ASH (Djurisic et al., 2023) works by pruning a large portion of an input sample's activation and lightly adjusting the remaining. The mentioned methods still rely on Energy or other output-based methods to detect OOD samples by observing the output space, rather than directly using feature patterns for OOD detection.

**Fusion methods** combine the information of features and outputs together to design OOD scores. Lee et al. (2018) calculate the Mahalanobis distance differences between the each layer's feature and the class prior feature as OOD scores. ViM (Wang et al., 2022a) leverages feature residuals and existing logits of the classifier to calculate OOD scores. HEAT (Lafon et al., 2023) use a hybrid Energy based model in the feature space to estimate the density of ID samples for OOD detection.

**Comparison with other methods.** Although React, ASH, and our approach all employ thresholds for feature processing, our fundamental ideas are entirely distinct. ReAct thinks that abnormally high activations in the fully connected layer can lead to the overconfidence, so it truncates all activations that are higher than the threshold. ASH is to preserve or even amplify the activations above the threshold in the penultimate layer of the classifier. In contrast, our method focuses on processing feature maps in the backbone and truncates all activations in intermediate stages that are lower than the threshold while weakening the other activations. Our method weakens a large number of OOD features that could potentially generate abnormal activations, preventing the foreground parts of OOD images from producing excessively high activations. Our method has sufficient network parameters following it to reprocess and restore features, allowing it to choose to retain salient (high) activations. More detailed analyses can be found in Appendix E.

## 3 PRELIMINARIES

Let $\mathcal{X}$ and $\mathcal{Y}^{\text{in}} = \{1, ..., K\}$ be the feature space and the ID label space. ID model $\mathbf{f}_\Theta$ is trained on the training data $\mathcal{D}_{\text{in}}^{\text{train}} = \{(\mathbf{x}^1, y^1), ..., (\mathbf{x}^n, y^n)\}$ and is deployed on the test data $\mathcal{D}^{\text{test}}$, which is not *independent and identically distributed* (i.i.d.) drawn from the distribution of $\mathcal{D}_{\text{in}}^{\text{train}}$ .

A well-known branch of OOD detection methods is to develop the post-hoc OOD detection (or inference-time OOD detection) methods (Huang et al., 2021; Liang et al., 2018; Liu et al., 2020; Hendrycks & Gimpel, 2017; Lee et al., 2018; Sun et al., 2021), where we often design an OOD score function in the inference time to recognize OOD data well. The key advantage of inference-time OOD detection methods is that the task performance on ID data will be unaffected since we do not change the ID model's parameters.

Given a threshold $\gamma$, an ID model $\mathbf{f}$ and a scoring function $S$, then OOD detection can be regarded as a binary classification problem:

$$G_\gamma(\mathbf{x}) = \begin{cases} \text{ID}, & \text{if } S(\mathbf{f}, \mathbf{x}) \geq \gamma \\ \text{OOD} , & \text{if } S(\mathbf{f}, \mathbf{x}) < \gamma \end{cases} \tag{1}$$

The performance of OOD detection depends on how to design a scoring function $S$ to make OOD data obtain lower scores while ID data obtain higher scores—thus, we can detect OOD data.

## 4    FEATURE MAP MATTERS IN OOD DETECTION

In *convolutional neural networks* (CNNs), the feature map refers to the output of the convolutional layer, which is generated by the pre-trained kernel convoluting on the input image. The feature map has richer spatial information than the feature vector in the classification head. Usually, CNNs first recognize an image's texture features (e.g., edges, shapes, and patterns). Then, CNNs combine the features into deep semantics and finally make predictions according to their activation amplitude.

For ID data, both its texture and semantic features should conform to the trained patterns of the model. For OOD data, on the other hand, its features generally coincidentally match a part of in-distribution texture patterns, thus making OOD data difficult to separate from ID data. Previous approaches focus on the classification stage of the network and directly use the classifier's output as the basis for OOD detection. However, the intermediate processes of the network in processing the features are also important, especially the activation-suppression differences (also known as foreground-background components), which contain rich clues and provide extra information for OOD detection, compared with the classification stage. In Fig. 1, we follow Peng et al. (2022) to visualize output feature maps of the backbone by heatmaps, and we can observe two phenomena: (1) ID features have higher activations and significant foreground component than OOD feature (see Figs. 1a and 1c). [1] (2) When we suppress the intermediate layer features, the ID data would behave more robustly than the OOD data (see Figs. 1b and 1d).

## 5    HOW TO USE FEATURE MAPS TO IMPLEMENT OOD DETECTION

In this section, we introduce the details of Feature Sim score and Threshold Activation module, and the whole pipeline of our OOD score is shown in Fig. 2.

### 5.1    FEATURE SIM SCORE

Feature Sim is an OOD detection method that only looks at the feature maps of the backbone network. Specifically, for input $\mathbf{x}$, we have the pre-trained feature extractor $\mathbf{f_\theta}$, and the feature map $\mathbf{f_\theta}(\mathbf{x}) \in \mathbb{R}^{C \times H \times W}$. We compute the mean value of activation on each channel:

$$\mu^{(c)} = \frac{1}{HW} \sum_{i=1}^{HW} \mathbf{f_\theta}(\mathbf{x})_i^{(c)}. \tag{2}$$

Then, FS is calculated by the mean value of the absolute deviation of the channel as the OOD score:

$$S(\mathbf{x}) = \frac{1}{CHW} \sum_{c=1}^{C} \sum_{i=1}^{HW} \left| \mathbf{f_\theta}(\mathbf{x})_i^{(c)} - \mu^{(c)} \right|. \tag{3}$$

Feature Sim measures the self-similarity of the feature maps, whose core idea is to compare the activation differences between foreground and background on the ID and OOD feature maps. In particular, Eq. (2) is equivalent to the GAP operator on on the spatial (height, width) axis. For each channel, a feature map with shape of (H, W) will be pooled as a single value. To be specific, the shape of f(x) is (B,C,H,W) and the shape of GAP(f(x)) is (B,C). Therefore, Eq. (3) can be simplified as:

$$S(\mathbf{x}) = \text{mean}(|\mathbf{f_\theta}(\mathbf{x}) - \text{GAP}(\mathbf{f_\theta}(\mathbf{x}))|), \tag{4}$$

where $\text{mean}(\cdot)$ represents the mean value calculated in $C$, $H$, and $W$ dimensions. The minus sign in Eq. (4) stands for tensor subtraction with the broadcast. The activation of the network for ID data is usually concentrated in the foreground, while the activation difference between foreground and background in OOD features is not as significant. Therefore, the ID data would obtain a larger Feature Sim score than the OOD data. Detailed theoretical analysis can be found in Section 7.

---

[1]Here, we provide additional clarification on the concepts of "foreground" and "background". "Foreground" does not merely refer to the image contents corresponding to its label (e.g., bedroom); rather, "foreground" pertains to the regions or objects that play a critical role in determining the sample labels (e.g.,furniture). The definition of "foreground" in our paper aligns with that of saliency detection [1], i.e., *saliency detection aims to locate the important parts of natural images which attract our attention* [2].

Table 1: OOD detection performance comparison with baselines. All methods are based on the same model trained on ID data only (ImageNet-1k). All values are percentages. ↑ indicates larger values are better and ↓ indicates smaller values are better. The shadow part represents our method.

| Method | iNaturalist | | SUN | | Places | | Textures | | Average | |
|---|---|---|---|---|---|---|---|---|---|---|
| | AUROC↑ | FPR95↓ | AUROC↑ | FPR95↓ | AUROC↑ | FPR95↓ | AUROC↑ | FPR95↓ | AUROC↑ | FPR95↓ |
| MSP | 88.42 | 52.72 | 81.75 | 68.54 | 80.63 | 71.58 | 80.46 | 66.15 | 82.82 | 64.75 |
| ODIN | 91.14 | 50.86 | 86.59 | 59.87 | 84.18 | 65.63 | 86.40 | 54.31 | 87.08 | 57.67 |
| Mahalanobis | 52.65 | 97.00 | 42.41 | 98.50 | 41.79 | 98.40 | 85.01 | 55.80 | 55.47 | 87.43 |
| Energy | 90.59 | 53.95 | 86.73 | 58.26 | 84.12 | 65.42 | 86.73 | 52.30 | 87.05 | 57.48 |
| GradNorm | 93.87 | 26.95 | 90.16 | 37.20 | 86.15 | 48.65 | 90.66 | 32.61 | 90.21 | 36.35 |
| ReAct | 96.22 | 20.38 | 94.20 | 24.20 | 91.58 | 33.85 | 89.90 | 47.30 | 92.95 | 34.43 |
| KNN [*] | 94.89 | 30.18 | 88.63 | 48.99 | 84.71 | 59.15 | 95.40 | 15.55 | 90.91 | 38.47 |
| DICE | 94.49 | 26.67 | 90.98 | 36.08 | 87.74 | 47.64 | 90.46 | 32.45 | 90.92 | 35.71 |
| ASH | 97.32 | 14.21 | 95.10 | 22.08 | 92.31 | 33.45 | 95.50 | 21.17 | 95.06 | 22.73 |
| FS+TA | 96.86 | 16.03 | 93.58 | 29.26 | 88.68 | 43.40 | 95.54 | 20.48 | 93.67 | 27.29 |
| FS+TA+ReAct | 97.52 | 12.41 | **95.66** | **21.21** | 92.52 | 34.41 | 96.23 | 18.01 | 95.48 | 21.51 |
| FS+TA+ASH | **98.15** | **9.60** | 95.59 | 21.52 | **93.05** | **32.43** | **97.09** | **14.70** | **95.97** | **19.56** |

[*] KNN reported here is actually KNN+, whose backbone is ResNet-50 pre-trained with contrastive loss.

## 5.2 THRESHOLD ACTIVATION

We propose the Threshold Activation module to increase the separability of ID and OOD data under Feature Sim scores. The design concept comes from two observations: (1) When we suppress feature maps in the middle layer, the ID data would behave more robustly than the OOD data (see Fig. 1). (2) The background noises of the image may confuse the ID and OOD data (Ming et al., 2022b).

Our Threshold Activation module is a plug-in module that acts on the network's middle-layer features (as shown in Fig. 2). It operates the feature maps $\mathbf{f}_{\boldsymbol{\xi}}(\mathbf{x})$ extracted from the middle layer with parameter $\boldsymbol{\xi}$ and puts them back for the rest of forward propagation. The module is defined as:

$$\mathbf{f}_{\boldsymbol{\xi}}^{'}(\mathbf{x}) = \mathrm{ReLU}(\mathbf{f}_{\boldsymbol{\xi}}(\mathbf{x}) - k) \tag{5}$$

where constant factor $k$ represents the threshold used to weaken the features of the middle layer. ReLU function is used to reactivate the weakened features and suppress the non-critical information. In this way, the feature maps are denoised, and the distance between ID and OOD data is broadened.

## 5.3 FUSION OF CLASSIFIER SPACE AND FEATURE MAP SPACE

Classifier-based methods and our feature map-based method discriminate OOD data from two different perspectives. So we consider the fusion of information from the two spaces to detect OOD data. Specifically, we combine Feature Sim score $S_{\mathrm{feat}}$ and classifier-based scores $S_{\mathrm{cls}}$ as:

$$S_{\mathrm{Fusion}} = S_{\mathrm{feat}} + \lambda S_{\mathrm{cls}}, \tag{6}$$

where $\lambda$ is a constant factor that controls the composite ratio of the two scores. We generally use $\lambda$ to scale $S_{\mathrm{cls}}$ so that the two scores are in the same order of magnitude, avoiding the absolute dominance of a single score in $S_{\mathrm{Fusion}}$.

## 6 EXPERIMENTS

### 6.1 EVALUATION ON LARGE-SCALE IMAGENET BENCHMARK

**Experiment setup.** We evaluate our method on the large-scale ImageNet benchmark (Huang & Li, 2021), where ImageNet-1k (Deng et al., 2009) is regarded as ID data. There are four OOD test datasets in the benchmark, which are sampled from iNaturalist (Horn et al., 2018), SUN (Xiao et al., 2010), Places (Zhou et al., 2018), and Textures (Cimpoi et al., 2014). Note that, there are no overlapping classes between ID datasets and OOD datasets (Huang & Li, 2021). We use ResNet50 (He et al., 2016) and DenseNet121 (Huang et al., 2016) pre-trained on ImageNet-1k in the following experiments. At test time, all images are resized to $224 \times 224$. We usually set $k = 0.2$ in Threshold Activation in the following experiments. The experiment details can be found in Appendix A.1. We use MSP (Hendrycks & Gimpel, 2017), ODIN (Liang et al., 2018), Mahalanobis (Lee et al., 2018), Energy (Liu et al., 2020), GradNorm (Huang et al., 2021) , ReAct (Sun et al., 2021), KNN (Sun et al., 2022), DICE (Sun & Li, 2022) and ASH (Djurisic et al., 2023) as the baselines for the comparison experiment and the details are shown in Appendix A.2.

**OOD detection performance comparison with baselines.** The results shown in Table 1 illustrate that the performance of our method FS+TA designed for feature map space is competitive with the

Table 2: Effect of Threshold Activation and Feature Sim on previous OOD detectors.

| OOD Detector | Threshold Activation | Feature Sim | iNaturalist | | SUN | | Places | | Textures | | Average | |
|---|---|---|---|---|---|---|---|---|---|---|---|---|
| | | | AUROC↑ | FPR95↓ | AUROC↑ | FPR95↓ | AUROC↑ | FPR95↓ | AUROC↑ | FPR95↓ | AUROC↑ | FPR95↓ |
| MSP | ✗ | ✗ | 88.42 | 52.78 | 81.75 | 68.57 | 80.63 | 71.60 | 80.46 | 66.17 | 82.82 | 64.78 |
| | ✓ | ✗ | 91.27 | 43.75 | 85.42 | 59.94 | 83.57 | 65.26 | 81.63 | 63.23 | 85.47 | 58.04 |
| | ✓ | ✓ | 97.62 | 11.36 | 94.62 | 24.24 | 91.01 | 36.92 | 96.60 | 16.61 | 94.96 | 22.28 |
| ODIN | ✗ | ✗ | 91.13 | 50.88 | 86.59 | 59.87 | 84.18 | 65.67 | 86.40 | 54.31 | 87.08 | 57.68 |
| | ✓ | ✗ | 91.83 | 50.13 | 89.47 | 52.27 | 86.95 | 60.38 | 86.35 | 56.38 | 88.65 | 54.79 |
| | ✓ | ✓ | 96.61 | 16.73 | 93.11 | 29.73 | 88.08 | 44.59 | 96.48 | 16.63 | 93.57 | 26.92 |
| Energy | ✗ | ✗ | 90.59 | 53.99 | 86.73 | 58.28 | 84.12 | 65.43 | 86.73 | 52.32 | 87.05 | 57.51 |
| | ✓ | ✗ | 90.46 | 58.65 | 89.69 | 51.53 | 86.95 | 60.87 | 86.74 | 54.49 | 88.46 | 56.38 |
| | ✓ | ✓ | 97.17 | 14.40 | 95.23 | 23.05 | 91.79 | 36.73 | 96.11 | 17.70 | 95.08 | 22.97 |
| GradNorm | ✗ | ✗ | 93.86 | 26.96 | 90.16 | 37.24 | 86.15 | 48.69 | 90.66 | 32.59 | 90.21 | 36.37 |
| | ✓ | ✗ | 96.10 | 20.02 | 94.97 | 23.52 | 91.69 | 35.09 | 90.67 | 34.02 | 93.36 | 28.16 |
| | ✓ | ✓ | 96.82 | 16.12 | 95.06 | 22.65 | 91.20 | 35.91 | 94.75 | 23.67 | 94.46 | 24.59 |
| ReAct | ✗ | ✗ | 96.22 | 20.38 | 94.20 | 24.20 | 91.58 | 33.85 | 89.90 | 47.30 | 92.95 | 34.43 |
| | ✓ | ✗ | 96.22 | 20.71 | 95.70 | 20.17 | 93.12 | 30.80 | 90.96 | 46.47 | 94.00 | 29.54 |
| | ✓ | ✓ | 97.52 | 12.41 | 95.66 | 21.21 | 92.52 | 34.41 | 96.23 | 18.01 | 95.48 | 21.51 |
| ASH | ✗ | ✗ | 97.32 | 14.21 | 95.10 | 22.08 | 92.31 | 33.45 | 95.50 | 21.17 | 95.06 | 22.73 |
| | ✓ | ✗ | 98.11 | 9.91 | 95.06 | 24.07 | 92.37 | 35.26 | 97.39 | 13.03 | 95.73 | 20.57 |
| | ✓ | ✓ | 98.15 | 9.60 | 95.59 | 21.52 | 93.05 | 32.43 | 97.09 | 14.70 | 95.97 | 19.56 |

classifier-output-based methods. This indicates that feature maps extracted from the backbone contain enough information to detect OOD data compared with the features or output from the classifier. Furthermore, our method FS+TA+ASH reaches state-of-the-art with 95.97% AUROC and 19.56% FPR95 on average across four OOD datasets. This shows that our FS+TA+ASH method overcomes the limitations of each method by fusing information from both feature space and classifier space, resulting in further performance improvement on each OOD dataset.

**Fusion of classifier space and feature map space methods.** We fuse Threshold Activation and Feature Sim with those previous OOD detection methods based on classifier outputs or features and the experimental results are shown in Table 2. The second row of each part in Table 2 suggests that Threshold Activation can be well compatible with other OOD scores and effectively improve the OOD detection performances of previous methods. This phenomenon implies that the possible reason for the confusion between OOD data and ID data is that the complex background of OOD data interferes with the classifier, thus making the model overconfident. This problem is also found by Ming et al. (2022b), which argues that the misclassification of some OOD samples is due to the model's overfitting to familiar backgrounds. Further, we conduct relevant experiments following the setting of the spurious correlation benchmark (Ming et al., 2022b) in Section 6.2.

The last row of each part in Table 2 shows that the combination of our proposed feature map-based and classifier-based methods has surprisingly effective results in improving the OOD detection performance of previous methods to the state-of-the-art. The performance gap between previous methods is narrowed a lot by our method. For example, the earliest and simplest method MSP can have ∼12% increase in AUROC and ∼42% decrease in FPR95. The main reason for the improvement is that our methods focus on the information of feature maps, and previous methods focus on the semantic information of the classifier's output. The two types of information are complementary so that the combination can achieve the highest performance.

**Threshold Activation and Feature Sim in different stages.** We now explore the effect of placement of Threshold Activation and Feature Sim, and the results are presented in Table 3. As shown in the first part of Table 3, the performance of Feature Sim increases as it is placed after the deeper stage, so we place Feature Sim after the $C_5$ stage. Differently, as shown in the last part of Table 3, the performance of Feature Sim first increases and then decreases as Threshold Activation is placed after the deeper stage, so we place Threshold Activation after the $C_4$ stage.

Our experiments reveal that the placement of Threshold Activation in non-output blocks $(C_2, C_3, C_4)$ enhances the separability of ID and OOD samples by distorting the representation of mid-layer features. The subsequent convolutional layers can then restore the activations related to ID patterns while keeping the OOD feature intensities low, as demonstrated in Fig. 1. However, placing Threshold Activation after the $C_5$ stage can be detrimental to OOD detection because there are no more convolutional layers to compensate for the distortion.

Table 3: Effect of Threshold Activation and Feature Sim in different stages. $C_i, i \in [2, 3, 4, 5]$ means the module is placed after $C_i$ stage.

| Threshold Activation | Feature Sim | iNaturalist | | SUN | | Places | | Textures | | Average | |
|---|---|---|---|---|---|---|---|---|---|---|---|
| | | AUROC↑ | FPR95↓ | AUROC↑ | FPR95↓ | AUROC↑ | FPR95↓ | AUROC↑ | FPR95↓ | AUROC↑ | FPR95↓ |
| ✗ | $C_2$ | 49.92 | 98.12 | 69.29 | 89.69 | 69.15 | 88.66 | 65.13 | 74.41 | 63.37 | 87.72 |
| | $C_3$ | 47.84 | 98.33 | 76.18 | 84.44 | 73.34 | 84.63 | 71.99 | 69.18 | 67.34 | 84.15 |
| | $C_4$ | 87.49 | 55.44 | **83.04** | 71.03 | **79.62** | 78.72 | 82.16 | 55.62 | 83.08 | 65.20 |
| | $C_5$ | **88.33** | **42.93** | 78.80 | 62.33 | 71.07 | **74.50** | **95.06** | **19.56** | **83.32** | **49.83** |
| $C_2$ | | 87.75 | 53.50 | 75.89 | 74.41 | 68.15 | 82.61 | 95.91 | 18.67 | 81.93 | 57.30 |
| $C_3$ | $C_5$ | 91.18 | 40.93 | 83.46 | 63.53 | 74.49 | 76.36 | **97.03** | **15.00** | 86.54 | 48.96 |
| $C_4$ | | **96.86** | **16.03** | **93.58** | **29.26** | **88.68** | **43.40** | 95.54 | 20.48 | **93.67** | **27.29** |
| $C_5$ | | 89.43 | 41.20 | 88.23 | 44.30 | 82.33 | 57.70 | 85.73 | 45.30 | 86.43 | 47.13 |

Table 4: OOD detecton performance on CIFAR benchmarks. The shown results are averaged over all OOD datasets and the full results are shown in Table 9. ↑ indicates larger values are better and ↓ indicates smaller values are better.

| Method | CIFAR10 | | CIFAR100 | |
|---|---|---|---|---|
| | AUROC↑ | FPR95↓ | AUROC↑ | FPR95↓ |
| MSP | 91.49 | 51.68 | 78.40 | 78.06 |
| ODIN | 93.07 | 35.55 | 79.97 | 75.15 |
| Energy | 93.11 | 35.26 | 80.05 | 74.80 |
| GradNorm | 84.66 | 51.10 | 70.49 | 72.87 |
| ReAct | 93.18 | 34.87 | 83.03 | 71.08 |
| KNN | 94.55 | 29.15 | 81.82 | 69.35 |
| DICE | 94.08 | 32.09 | 76.50 | 79.09 |
| FS+TA | **96.30** | **20.24** | **83.75** | **56.31** |

Table 5: OOD detection performance of models trained on Waterbirds. We choose r=0.9 for Waterbirds. The results of Non-spurious OOD are averaged over 3 OOD test datasets: SVHN, iSUN, LSUN (see details in Appendix B.6).

| Method | Spurious OOD | | Non-spurious OOD | |
|---|---|---|---|---|
| | AUROC↑ | FPR95↓ | AUROC↑ | FPR95↓ |
| MSP | 74.38 | 82.90 | 84.55 | 70.93 |
| ODIN | 74.38 | 82.90 | 84.55 | 70.93 |
| Energy | 75.39 | 81.30 | 86.20 | 61.80 |
| GradNorm | 49.97 | 80.55 | 71.32 | 59.32 |
| ReAct | 80.65 | 84.60 | 87.12 | 60.48 |
| KNN | 65.21 | 73.40 | 94.86 | 17.07 |
| DICE | 80.19 | 60.55 | 86.21 | 49.32 |
| FS+TA | **81.72** | **48.80** | **97.40** | **15.78** |

## 6.2 FURTHER EXPLORATION AND DISCUSSION

**Evaluation on CIFAR Benchmarks.** Following the setup in (Sun et al., 2022), we use CIFAR10 and CIFAR100 datasets (Krizhevsky, 2009) as ID data and train ResNet18 (He et al., 2016) models on them respectively for the following experiments. There are five OOD test datasets in CIFAR benchmarks: SVHN (Netzer et al., 2011), LSUN-Crop (Yu et al., 2015), iSUN (Xu et al., 2015), Places (Zhou et al., 2018), and Textures (Cimpoi et al., 2014). We usually set $k = 0.2$ for CIFAR10 and $k = 0.15$ for CIFAR100 in Threshold Activation. At test time, all images are of size $32 \times 32$. The training details can be found in Appendix A.1. Comparison with competitive OOD detection methods. To further validate the generalization of our approach, we evaluate our method on the CIFAR benchmark and conduct comparative experiments. Table 4 reports the performances of our approach and shows that our approach significantly outperforms existing baselines.

**Spurious correlation.** Ming et al. (2022b) provide a new OOD detection formalization and divide OOD data into two types: (1) spurious OOD - test samples whose semantics do not overlap with ID data but whose image background is similar to ID data; (2) non-spurious OOD - test samples whose semantics and image background do not overlap with ID data. Ming et al. (2022b) argue that spurious correlation in the training data poses a significant challenge to OOD detection. Therefore, we follow the setup of Ming et al. (2022b) for our experiments, where Waterbirds (Sagawa et al., 2019) is used as the ID dataset to train a ResNet18, subsets of Places are used as the Spurious OOD dataset, and SVHN, iSUN, LSUN are used as the Non-spurious OOD dataset. The results shown in Table 5 illustrate that our approach can effectively solve the performance degradation caused by spurious correlation. The main reason for this phenomenon is that our method designs the OOD score function Feature Sim by analyzing the relationship between foreground and background from the feature map level, and Threshold Activation can suppress the activation of background features, so it can effectively solve the problem caused by similar background in both ID and OOD data.

**Detecting OOD samples in more tasks.** Since our method only looks at feature maps and does not rely on classifiers, it can be adapted to more computer vision tasks. We evaluate our method on object detection and semantic segmentation. The experimental results are shown in Table 6. Concretely, we extract backbones from the task-specific networks for feature map-based OOD detection. The implementation details can be found in Appendix A.1. In these tasks, previous methods (Hendrycks & Gimpel,

Table 6: Results on object detection and semantic segmentation. The values are averaged over 4 OOD datasets. The detailed results are shown in Table 11.

| ID Task | AUROC↑ | FPR95↓ |
|---|---|---|
| Object Detection | 85.16 | 62.88 |
| Semantic Segmentation | 88.61 | 45.75 |

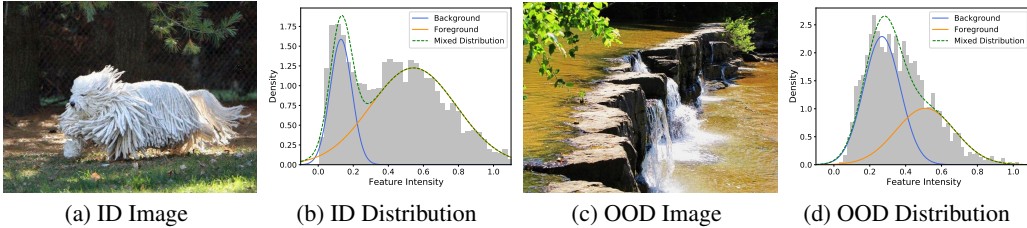

| (a) ID Image | (b) ID Distribution | (c) OOD Image | (d) OOD Distribution |

Figure 3: Pixel-wise feature map intensity distribution of ID and OOD data. The gray bar represents the histogram of real data. The blue and orange solid lines represent the EM-estimated background and foreground components of the GMM model. The green dashed line represents the mixed distribution. The distance between the foreground and background of ID features is larger than that of OOD features, indicating that the foreground is more salient in ID data. More visualization results can be found in Appendix C.2.

2017; Du et al., 2022; Chan et al., 2021) can only perform OOD detection at the instance or pixel level and lacks a global view. Therefore, they focus on determining whether each object in the picture belongs to the ID and it is impossible to determine whether an input image is ID. Our method can accomplish image-level OOD detection without introducing additional parameters and designs.

**More experiments.** We discuss the ablation study of threshold $k$ and the computation cost in Appendix B.2. We evaluate our method on DenseNet-121 in Appendix B.8 and near OOD benchmark in Appendix B.9. The ablation study of score form of Feature Sim can be found and Appendix B.10.

# 7 UNDERSTANDING FEATURE SIM FROM THE PERSPECTIVE OF GMM

To better understand the effectiveness of our approach, inspired by Ming et al. (2022b), we model the foreground and background feature intensities of an image as one-dimensional independent Gaussian distributions so that feature map activation histograms can be modeled by *Gaussian mixture model* (GMM). These modeling assumptions are based on the activation statistics observed on ID and OOD images (see Fig. 3). As mentioned above, measuring the distance between foreground and background components can effectively identify OOD data. However, decomposing or fitting feature maps' foreground and background components are computationally intensive. In the following analysis, we prove that the proposed Feature Sim score positively correlates with the distance between foreground and background distributions and is effective for feature map-based OOD detection.

*Setup.* For an input image, we denote the feature maps output by the backbone $\mathbf{f}_{\boldsymbol{\theta}}$ as $\mathbf{f}_{\boldsymbol{\theta}}(\mathbf{x}) \in \mathbb{R}^{C \times H \times W}$. The semantic information in each channel of the feature map is different, so we assume that the distribution of each channel is independent to simplify the theoretical modeling. For each channel $c$, we flatten the feature map $\mathbf{f}_{\boldsymbol{\theta}}^{(c)}(\mathbf{x})$ in spatial dimension and note it as $h^{(c)} \in \mathbb{R}^{HW}$, whose mean and standard deviation are $\mu^{(c)}$ and $\sigma^{(c)}$, respectively. Note that $h^{(c)}$ is one-dimensional, and the histogram in Fig. 3 shows its distribution. In the following formulas, we have omitted the superscript $(c)$ because of the independence.

Following Ming et al. (2022b), we assume both the foreground intensities $f$ and background intensities $g$ in $h$ are drawn from two Gaussian distributions: $f \sim N(\mu_1, \sigma_1^2), g \sim N(\mu_2, \sigma_2^2)$. Then we define the estimated feature distribution $\hat{h}$ by the weighted sum of $f$ and $g$: $\hat{h} = w \cdot f + (1 - w) \cdot g$, where $w \in [0, 1]$ is the mixture weight of foreground and background distributions. Since we define the foreground as the component with higher activation in the feature map, it is natural that $\mu_1 \geq \mu_2$. Thus, we could model feature maps $h$ by GMM and the estimated probability density function is:

$$p(\hat{h}) = w \cdot N(\mu_1, \sigma_1^2) + (1 - w) \cdot N(\mu_2, \sigma_2^2). \tag{7}$$

*Remark* 7.1. The mean of the estimated GMM is aligned with the mean of the real distribution.

In practice, the EM algorithm (McLachlan & Krishnan, 2007) is a typical method for solving GMM parameters and can be used to fit the feature map intensities from the backbone. Then the following equation holds and the proof can be found in Appendix D.2:

$$\mu = w \cdot \mu_1 + (1 - w) \cdot \mu_2. \tag{8}$$

We apply Feature Sim score to estimated distribution $\hat{h}$, which can be treated as applying separately on the foreground and background components. So we have $f^* = |f - \mu|, g^* = |g - \mu|$, where $f^*$

and $g^*$ are Folded Normal Distribution (Leone et al., 1961) with the expectations as:

$$\begin{aligned}
\mathbb{E}(f^*) &= (\mu_1 - \mu)[1 - 2\Phi(-(\mu_1 - \mu)/\sigma_1)] + 2\sigma_1\phi((\mu_1 - \mu)/\sigma_1), \\
\mathbb{E}(g^*) &= (\mu_2 - \mu)[1 - 2\Phi(-(\mu_2 - \mu)/\sigma_2)] + 2\sigma_2\phi((\mu_2 - \mu)/\sigma_2),
\end{aligned} \tag{9}$$

where $\phi$ and $\Phi$ represent the standard normal distribution's probability density and cumulative distribution function. The expectation of $|\hat{h} - \mu|$ is shown below and the proof is shown in Appendix D.3:

$$\mathbb{E}(|\hat{h} - \mu|) = w \cdot \mathbb{E}(f^*) + (1 - w) \cdot \mathbb{E}(g^*). \tag{10}$$

Assume that $d = \mu_1 - \mu_2 \geq 0$. Based on Eq. (8) and Eq. (10), we can simplify our FS score as:

$$\begin{aligned}
s = {}& 2w(w - 1) \cdot d\Phi((w - 1)d/\sigma_1) + 2w(1 - w) \cdot d\Phi(wd/\sigma_2) \\
& + 2w \cdot \sigma_1\phi((1 - w)d/\sigma_1) + 2(1 - w) \cdot \sigma_2\phi(wd/\sigma_2)
\end{aligned} \tag{11}$$

*Remark* 7.2. Feature Sim is positively related to the difference between the foreground and background components. ID data have higher Feature Sim scores than OOD data.

We compute the partial derivative with respect to $d$:

$$\frac{\partial s}{\partial d} = 2w(1 - w)(\Phi(wd/\sigma_2) - \Phi((w - 1)d/\sigma_1)) \geq 0. \tag{12}$$

The Eq. (12) holds because every term in it is non-negative. Particularly, $wd/\sigma_2$ is a non-negative logit and $(w - 1)d/\sigma_1$ is a non-positive logit, so $\Phi(wd/\sigma_2) - \Phi((w - 1)d/\sigma_1) \geq 0$.

Thus, the expectation $\mathbb{E}(|x - \mu|)$ is a monotonically increasing function with respect to $d$, so our Feature Sim score is positively related to $d$. As the difference between the background and foreground features increases, our Feature Sim score increases as well.

We also design an OOD score (called GMM score) through the GMM-estimated distributions to directly measure the background-foreground differences and the results are shown in Table 17. In detail, we design a GMM-based score $S_{GMM} = (\mu_f - \mu_g) \cdot \frac{w_f}{w_g}$, which takes channel-wised foreground-background statistics $\mu_f, \mu_g$ and is scaled by the proportion of the two components. The scale factor is used because a sample with a higher foreground weight is more likely to be the ID. The performance on the ImageNet benchmark is 90.83% AUROC and 35.08 % FPR95 and is close to our proposed method, illustrating that our theory explains our method properly. We think the performance gap is mainly caused by the estimation errors of GMM.

Compared with the GMM score, Feature Sim score measures the foreground and background distance without introducing a heavy computational burden for estimating the statistics of foreground $f$ and background $g$. Based on our empirical observations in Fig. 3, ID data have more significant foreground activations than OOD data, i.e., $d_{ID} > d_{OOD}$. Therefore, Feature Sim scores for ID data are significantly greater than those for OOD data.

## 8 CONCLUSION

In this paper, we propose a new perspective to discriminate OOD data by only looking at feature maps, without relying on the model's classifier. Deep feature maps contain both semantic information and foreground-background spatial relationships. First, Feature Sim score designs the OOD score based on the mean absolute deviation of the feature maps, which detects OOD samples through the differences between foreground and background components. Then, Threshold Activation module suppresses the feature activations in the middle layer of backbone, which not only increases the difference between ID and OOD, but also reduces the effect of background noise on OOD detection. Finally, we fuse the knowledge in feature map space and output space. The proposed fusion score can fully exploit the clues of OOD samples in the two spaces and discriminate OOD data more comprehensively. Extensive experiments show that our methods FS+TA and FS+TA+ASH achieve the state-of-the-art on various benchmarks.

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

# A More Experimental Details

## A.1 Model Hyperparameters

Our code is implemented with Python 3.9 and PyTorch 1.9. We conduct all the experiments on NVIDIA GeForce RTX 3090Ti GPUs.

We use checkpoints of ResNet18, ResNet50, and DenseNet121 provided by *mmclassification*[2] (Contributors, 2020a). The training details are shown below:

- As for ResNet18, we set the batch size as 16 and train the model on the CIFAR datasets for 200 epochs. We use SGD optimizer with momentum as 0.9 and weight decays as 0.0005. The start learning rate is 0.1 and decays at epochs 60, 120, and 180.

- As for ResNet50, we set the batch size as 128 and train the model on the ImageNet-1k for 200 epochs. We use SGD optimizer with momentum as 0.9 and weight decays as 5e-3. The start learning rate is 0.1 and decays at epochs 60, 120, and 180.

- As for DenseNet121, we set the batch size as 256 and train the model on the ImageNet-1k for 90 epochs. We use SGD optimizer with momentum as 0.9 and weight decays as 0.0001. The start learning rate is 0.1 and decays at epochs 30, 60, and 90.

We use *mmdetection*[3] (Chen et al., 2019) to train Faster-RCNN (Ren et al., 2017) on MS COCO (Lin et al., 2014). The detector uses ImageNet-pre-trained ResNet50 as the backbone and uses feature pyramid networks (FPN) for multi-scale enhancement. The network is trained on the COCO dataset for 12 epochs with SGD as an optimizer. The batch size and learning rate are set to 16 and 0.01, respectively. Other settings are consistent with the original Faster R-CNN implementation.

We use *mmsegmentation*[4] (Contributors, 2020b) to train FCN (Shelhamer et al., 2017) on ADE20K (Zhou et al., 2019). The network uses ImageNet-pre-trained ResNet50 as the backbone and is trained on the ADE20K dataset for 80,000 iterations with SGD as the optimizer. The batch size and learning rate are set to 16 and 0.01, respectively. Other settings are consistent with the original FCN implementation.

Then, we extract the backbone of ResNet50 from Faster-RCNN and FCN for OOD detection. We set $k = 0.2$ for Faster-RCNN and $k = 0.15$ for FCN in Threshold Activation.

The fuse hyperparameter $\lambda$ is 0.1 for MSP, 0.016 for Energy, 1300 for ODIN, 0.4 for GradNorm and 0.028 for ReAct.

The introduction of $\lambda$ is the most intuitive approach as it normalizes the variation ranges of the two methods to the same interval, making the fusion weights approximately equal. It is important to note that this normalization is independent of the specific data distribution and only depends on the characteristics of the two methods involved in the fusion. Hence, there is no issue of test set leakage.

Additionally, $\lambda$ is not a meticulously searched parameter; we simply adjust the dynamic range of the two scores to be of the same order of magnitude, allowing the fused score function to adequately incorporate opinions from both convolutional features and the classifier, rather than having one dominate absolutely. In fact, $\lambda$ is robust to variations over a wide range as shown in Fig. 4.

As for the position of FS and TA, we think FS must be placed after the C5 stage. The ablation study in Table 3 is just to demonstrate the correctness of our foreground-background theory, showing that FS can be effective at multiple positions. However, placing FS after C5 stage truly leverages the network's full capability to process and analyze images. Indeed, there are multiple choices for TA, but its performance varies significantly when placed after each stage; when it is positioned after C4, the performance of FS+TA is much higher than at other positions. We follow the approach of DICE and use Gaussian noise images as a validation set to select the position for TA.

---

[2] https://github.com/open-mmlab/mmclassification
[3] https://github.com/open-mmlab/mmdetection
[4] https://github.com/open-mmlab/mmsegmentation

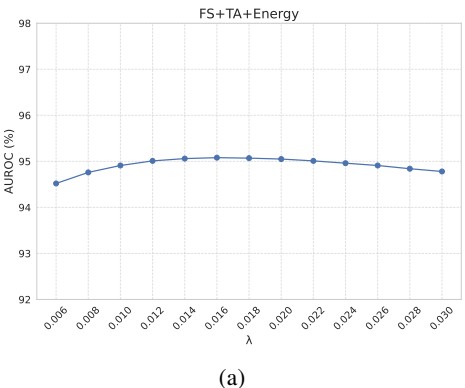 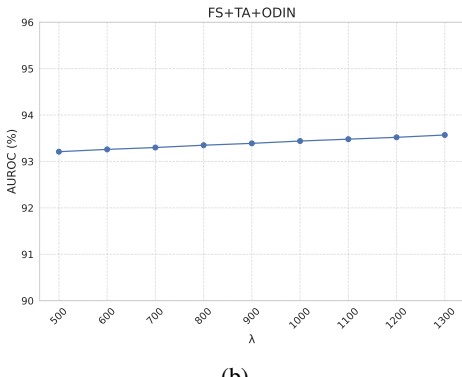

(a) (b)

Figure 4: AUROC changes with respect to $\lambda$. We can notice that the OOD performances of FS+TA+Energy and FS+TA+ODIN are stable when $\lambda$ varies within an order of magnitude.

## A.2 Descriptions of Baseline Methods

For the convenience of the reader, we summarise in detail some common techniques for OOD detection. All methods calculate OOD scores on a neural network trained using only in-distribution data in the inference time.

- MSP (Hendrycks & Gimpel, 2017) uses the maximum probability of the softmax function as the OOD score. They argue that ID data would have a higher maximum probability than OOD data.

- ODIN (Liang et al., 2018) uses input perturbation strategy and adds temperature scaling to the softmax function. In all experiments, we set the temperature scaling parameter $T = 1000$.

- Mahalanobis (Lee et al., 2018) uses multivariate Gaussian distributions to model class-conditional distributions of each layer and proposes Mahalanobis distance-based scores for OOD detection.

- Energy (Liu et al., 2020) uses the negative energy function as the OOD score, and the input is logit outputs of the classifier's penultimate layer.

- GradNorm (Huang et al., 2021) uses the vector norm of the gradients obtained by back-propagating the KL divergence between the softmax output and the uniform probability distribution as OOD scores.

- ReAct (Sun et al., 2021) clips noisy activations of the classifier's penultimate layer and uses Energy score to calculate OOD scores. The threshold $c$ is set as the 90-percentile of ID activations for ResNet50 and ResNet18 and 1.5 for DenseNet121.

- KNN (Sun et al., 2022) calculates the $k$-th nearest neighbor (KNN) distance between input embeddings as OOD scores. We use $k = 1000$ in all experiments.

- DICE (Sun & Li, 2022) selectively uses top-$k$ weights of the classifier's penultimate layer and calculates OOD scores through Energy score. The top-$k$ weights are calculated by the sparsity parameter $p$, and we use $p = 0.7$ in all experiments.

- ASH (Djurisic et al., 2023) ASH removes most of the activations of the fully-connected layers of the classifier and applies shaping to the rest.

## B More Experimental Results

### B.1 Evaluation Metrics

We use two common metrics to evaluate OOD detection methods (Huang et al., 2021): the false positive rate that OOD data are classified as ID data when 95% of ID data are correctly classified (FPR95) (Provost et al., 1998) and the *area under the receiver operating characteristic curve* (AUROC) (Huang et al., 2021).

### B.2 ABLATION STUDY ON THRESHOLD $k$.

We explore the effect of threshold $k$ in Threshold Activation. In Fig. 5, we summarize the OOD performance and ID accuracy for ResNet50 trained on ImageNet-1k, where we vary $k$ from 0 to 0.5. As the threshold $k$ increases, the ID accuracy first decreases slowly and then rapidly decreases to near 0, while the OOD performance shows an ascending and then descending trend. The main reason for this phenomenon is that when the threshold $k$ is small, the difference between ID and OOD data is not large enough, so as $k$ increases, the OOD detection performance gradually increases; when the threshold $k$ is large, Threshold Activation would weaken the representation of ID data, thus making the discrimination errors of ID data, which in turn is detrimental to OOD detection. Therefore, we use the inflection point $k = 0.2$ as the optimal threshold in Threshold Activation.

Although we use the ID classification accuracy as the metric to determine a suitable $k$, we do not intend to use the TA module for the ID classification task. It is important to note that our method does not affect the classification performance and only incurs a computational overhead of 7%, as evidenced in Appendix B.4. Specifically, we utilize the original features without the TA module for the ID classification task and obtain a copy of the mid-layer features at the TA insertion point for OOD detection. This ensures that the ID classification accuracy remains unaffected.

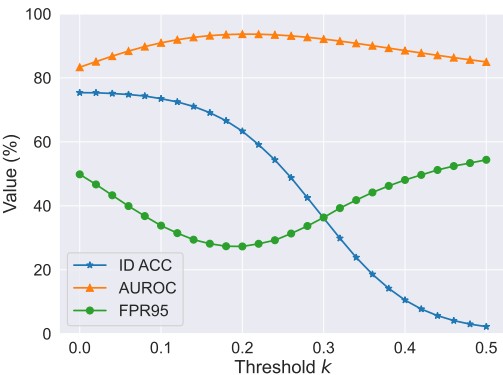

Figure 5: Effect of threshold $k$ in Threshold Activation. Detailed results are shown in Appendix B.3.

### B.3 ID ACCURACY AND OOD DETECTION PERFORMANCE WITH THRESHOLD ACTIVATION

Although threshold activation is a plug-and-play module and does not change the model parameters, the ID classification accuracy after inserting the threshold activation module is also worthy of attention. For this reason, we insert threshold activation after $C_4$ stage to observe the effect of different threshold values on ID classification accuracy and OOD detection performance. The results are shown in Table 7.

Table 7: Effect of different threshold values on ID classification accuracy and OOD detection performance.

| k | 0.00 | 0.02 | 0.04 | 0.06 | 0.08 | 0.10 | 0.12 | 0.14 | 0.16 | 0.18 | 0.20 | 0.22 | 0.24 | 0.26 | 0.28 | 0.30 | 0.32 | 0.34 | 0.36 | 0.38 | 0.40 | 0.42 | 0.44 | 0.46 | 0.48 | 0.50 |
|---|---|---|---|---|---|---|---|---|---|---|---|---|---|---|---|---|---|---|---|---|---|---|---|---|---|---|
| AUROC (%) | 83.32 | 85.06 | 86.79 | 88.37 | 89.77 | 90.95 | 91.92 | 92.66 | 93.19 | 93.52 | 93.67 | 93.64 | 93.45 | 93.13 | 92.68 | 92.13 | 91.49 | 90.79 | 90.05 | 89.29 | 88.53 | 87.78 | 87.04 | 86.33 | 85.64 | 84.98 |
| FPR95 (%) | 49.83 | 46.68 | 43.28 | 39.92 | 36.78 | 33.81 | 31.46 | 29.40 | 28.15 | 27.35 | 27.29 | 28.12 | 29.23 | 31.33 | 33.68 | 36.35 | 39.28 | 41.78 | 44.17 | 46.22 | 48.08 | 49.65 | 51.19 | 52.42 | 53.36 | 54.38 |
| ID ACC (%) | 75.36 | 75.33 | 75.11 | 74.79 | 74.32 | 73.48 | 72.47 | 71.05 | 69.11 | 66.59 | 63.32 | 59.12 | 54.34 | 48.76 | 42.51 | 36.16 | 29.85 | 23.85 | 18.59 | 14.17 | 10.52 | 7.72 | 5.65 | 4.11 | 3.00 | 2.23 |

### B.4 ADDITIONAL COMPUTATIONAL BURDEN IMPOSED BY FEATURE SIM AND THRESHOLD ACTIVATION

As shown in Fig. 2, the proposed Threshold Activation (TA) module modifies the middle-layer features, resulting in detached forward paths for ID classification and OOD detection tasks. Concretely, the input data should go through the shallow convolutional layers and becomes a middle-layer feature. The TA-modified features are dedicated to OOD detection and may reduce the accuracy of the original ID classification task (as discussed in Appendix B.3). So, in order to ensure the consistency of the original ID task results, our proposed method equivalents to introduce an extra

group of deep convolutional layers with sharing weights to the original network. And the forward pass of middle-layer features cannot be merged because of the nonlinearity of the network. This brings an additional computational burden.

Table 8 reports the timing results normalized by the ID task forward time. In keeping with the optimal design, the Threshold Activation module is placed after the $C_4$ stage. This brings an extra computational burden of double-forwarding in the $C_5$ stage for the features with or without Threshold Activation. Our method introduces about 7% extra forwarding time, which we think is acceptable because of the performance improvement.

Table 8: The timing results normalized by the time of one forward pass in the ID task.

|  | ID Task | MSP | Energy | Mahalanobis | Feature Sim and Threshold Activation |
|---|---|---|---|---|---|
| Normalized Time | 1.00 | 1.00 | 1.00 | 1.21 | 1.07 |

## B.5 DEATAILED RESULTS IN CIFAR BENCHMARK

The detailed results are shown in Table 9.

Table 9: OOD detecton performance on CIFAR benchmarks. All values are percentages. ↑ indicates larger values are better and ↓ indicates smaller values are better. The bold represents the best performance. the shadow part represents our method.

| ID Dataset | Method | SVHN AUROC↑ | SVHN FPR95↓ | LSUN AUROC↑ | LSUN FPR95↓ | iSUN AUROC↑ | iSUN FPR95↓ | Places AUROC↑ | Places FPR95↓ | Textures AUROC↑ | Textures FPR95↓ | Average AUROC↑ | Average FPR95↓ |
|---|---|---|---|---|---|---|---|---|---|---|---|---|---|
| CIFAR10 | MSP | 90.82 | 57.24 | 94.54 | 39.05 | 93.86 | 43.52 | 89.71 | 56.56 | 88.54 | 62.04 | 91.49 | 51.68 |
|  | ODIN | 91.78 | 45.32 | 98.17 | 10.34 | 95.76 | 25.73 | 91.57 | 40.97 | 88.08 | 55.37 | 93.07 | 35.55 |
|  | Energy | 91.79 | 45.38 | 98.31 | 9.51 | 95.81 | 25.17 | 91.63 | 40.65 | 88.03 | 55.59 | 93.11 | 35.26 |
|  | GradNorm | 81.93 | 62.44 | 98.41 | 8.68 | 86.86 | 53.64 | 82.58 | 62.80 | 73.53 | 67.93 | 84.66 | 51.10 |
|  | ReAct | 92.28 | 42.05 | 98.14 | 10.63 | 95.71 | 25.60 | 91.01 | 41.74 | 88.74 | 54.31 | 93.18 | 34.87 |
|  | KNN | 95.48 | 27.98 | 96.84 | 18.5 | 95.52 | 24.68 | 89.93 | 47.84 | 94.96 | 26.74 | 94.55 | 29.15 |
|  | DICE | 91.57 | 45.52 | 99.83 | 0.77 | 94.42 | 34.65 | 92.16 | 39.96 | 92.43 | 39.59 | 94.08 | 32.10 |
|  | FS+TA | **96.38** | **21.04** | **99.92** | **0.11** | **96.05** | **24.05** | **93.08** | **37.16** | **96.06** | **18.83** | **96.30** | **20.24** |
| CIFAR100 | MSP | 81.31 | 76.60 | 83.92 | 68.16 | 82.03 | 71.34 | 70.77 | 87.93 | 73.96 | 86.28 | 78.40 | 78.06 |
|  | ODIN | 83.39 | 73.92 | 87.63 | 58.38 | 85.05 | 68.88 | 69.41 | 88.28 | 74.35 | 86.31 | 79.97 | 75.15 |
|  | Energy | 83.46 | 74.25 | 88.30 | 54.51 | 85.22 | 70.21 | 69.09 | 88.44 | 74.17 | 86.58 | 80.05 | 74.80 |
|  | GradNorm | 73.96 | 75.67 | 92.28 | 38.21 | 69.96 | 75.53 | 55.66 | 91.33 | 60.57 | 83.60 | 70.49 | 72.87 |
|  | ReAct | 84.63 | 73.89 | 87.06 | 58.65 | **89.26** | **56.57** | 73.42 | 85.34 | 80.79 | 80.96 | 83.03 | 71.08 |
|  | KNN | 89.16 | 56.69 | 75.93 | 87.48 | 86.81 | 57.23 | 73.62 | 81.85 | 83.56 | 63.49 | 81.82 | 69.35 |
|  | DICE | 81.04 | 78.64 | 91.66 | 44.14 | 76.53 | 88.68 | 64.55 | 95.69 | 68.72 | 88.30 | 76.50 | 79.09 |
|  | FS+TA | **89.67** | **48.27** | **95.89** | **23.90** | 73.04 | 90.62 | **75.42** | **77.64** | **84.73** | **41.13** | **83.75** | **56.31** |

## B.6 SPURIOUS CORRELATION

Ming et al. (2022b) divide OOD data into two types: spurious OOD and non-spurious OOD, and argue that spurious correlation in the training data poses a significant challenge to OOD detection. Therefore, we follow the setup of Ming et al. (2022b) for our experiments. We train a ResNet18 on Waterbirds Sagawa et al. (2019) for 30 epochs with SGD optimizer. Subsets of Places related to water and land are used as the Spurious OOD dataset, and SVHN, iSUN, LSUN are used as the Non-spurious OOD dataset. The results are shown in Table 10. It is obvious that our approach achieves significant performance improvements on each dataset.

## B.7 DETECTING OOD SAMPLES IN MORE TASKS

Our approach involves conducting OOD detection directly before object detection and segmentation tasks, rejecting obvious OOD samples upfront. The detailed results are shown in Table 11.

Firstly, we believe preemptively discarding clear OOD samples before focusing on granular tasks like detection and segmentation is beneficial. This is because task-specific OOD detection (at object or pixel level) is more costly and requires custom designs for each visual task. For example, if a network's task is to detect apples in orchard images and a camera consistently captures the ground due to being loose, identifying this OOD image before performing specific detection tasks can significantly reduce risks and costs associated with OOD samples.

Table 10: OOD detection performance of models trained on Waterbirds. We choose r=0.9 for Waterbirds.

| Method | SVHN | | LSUN | | iSUN | | Spurious OOD | |
|---|---|---|---|---|---|---|---|---|
| | AUROC↑ | FPR95↓ | AUROC↑ | FPR95↓ | AUROC↑ | FPR95↓ | AUROC↑ | FPR95↓ |
| MSP | 77.28 | 79.85 | 91.31 | 60.05 | 85.04 | 72.90 | 74.38 | 82.90 |
| ODIN | 77.28 | 79.85 | 91.31 | 60.05 | 85.04 | 72.90 | 74.38 | 82.90 |
| Energy | 80.40 | 71.75 | 94.07 | 36.65 | 84.14 | 77.00 | 75.39 | 81.30 |
| KNN | 90.10 | 31.50 | 98.66 | 4.50 | **95.83** | **15.20** | 65.21 | 73.40 |
| DICE | 88.04 | 43.00 | 92.95 | 29.05 | 77.64 | 75.90 | 80.19 | 60.55 |
| FS+TA | **98.83** | **4.90** | **99.46** | **2.25** | 93.90 | 40.20 | **81.72** | **48.80** |

Secondly, the current domain of OOD detection lacks a unified Image-Level OOD benchmark and baselines tailored for various visual tasks. However, our method demonstrates the potential to consistently detect OOD samples across different tasks by assessing whether a network's feature extractor is "familiar" with an input sample. While we have only tested on Object Detection and Semantic Segmentation tasks, we believe our approach can generalize to a wider range of tasks, such as depth estimation, pose recognition, and pedestrian re-identification. To our knowledge, there are no specialized OOD detection methods for these tasks yet, which is why we describe FS+TA as a unified method.

We also use a simple average of existing pixel-wise OOD scores as the image-level OOD score (baseline) for the segmentation task and the results are shown in Table 12.

Table 11: Results on object detection and semantic segmentation.

| Task | Model | Backbone | ID Dataset | iNaturalist | | SUN | | Places | | Textures | | Average | |
|---|---|---|---|---|---|---|---|---|---|---|---|---|---|
| | | | | AUROC↑ | FPR95↓ | AUROC↑ | FPR95↓ | AUROC↑ | FPR95↓ | AUROC↑ | FPR95↓ | AUROC↑ | FPR95↓ |
| Object Detection | Faster R-CNN | ResNet50 | COCO | 78.08 | 80.80 | 91.60 | 46.40 | 88.06 | 58.20 | 82.89 | 66.10 | 85.16 | 62.88 |
| Image Segmentation | FCN | ResNet50 | ADE20K | 95.44 | 25.25 | 85.31 | 63.00 | 78.61 | 72.25 | 95.08 | 22.50 | 88.61 | 45.75 |

Table 12: Performance comparison of OOD detection in the segmentation task. The ID dataset is ADE20k and the model is FCN.

| Method | iNaturalist | | SUN | | Places | | Textures | | Average | |
|---|---|---|---|---|---|---|---|---|---|---|
| | AUROC↑ | FPR95↓ | AUROC↑ | FPR95↓ | AUROC↑ | FPR95↓ | AUROC↑ | FPR95↓ | AUROC↑ | FPR95↓ |
| MSP | 82.12 | 66.25 | 62.90 | 89.25 | 64.44 | 91.00 | 83.80 | 50.75 | 73.32 | 74.31 |
| NegLabel | **95.44** | **25.25** | **85.31** | **63.00** | **78.61** | **72.25** | **95.08** | **22.50** | **88.61** | **45.75** |

## B.8 EVALUATION ON DENSENET121

We also evaluate our method on DenseNet121, and the results are shown in Table 13. It can be seen that our method can still be comparable to the existing baselines and achieve state-of-the-art performance on average across four OOD datasets.

## B.9 NEAR OOD BENCHMARK

We use ResNet18 and ResNet50 on different near OOD benchmarks. The results in Table 14 show that our method FS+TA+Energy outperforms the baselines. Generally speaking, ReAct represents ReAct+Energy, and DICE represents the combination of DICE and Energy. We find that the near OOD benchmark (Fort et al., 2021) is a challenging setting, especially when CIFAR100 is used as the ID and CIFAR10 is used as the OOD, as all methods are unable to separate the ID and OOD very well.

Table 13: OOD detection performance on DenseNet. All methods are based on the same model trained on ID data only (ImageNet-1k). All values are percentages. ↑ indicates larger values are better and ↓ indicates smaller values are better. The bold represents the best performance. the shadow part represents our method.

| Method | iNaturalist | | SUN | | Places | | Textures | | Average | |
|---|---|---|---|---|---|---|---|---|---|---|
| | AUROC↑ | FPR95↓ | AUROC↑ | FPR95↓ | AUROC↑ | FPR95↓ | AUROC↑ | FPR95↓ | AUROC↑ | FPR95↓ |
| MSP | 89.05 | 49.25 | 81.54 | 67.04 | 81.05 | 69.26 | 79.19 | 67.06 | 82.71 | 63.15 |
| ODIN | 92.81 | 39.58 | 87.03 | 54.78 | 85.05 | 59.66 | 85.01 | 54.66 | 87.48 | 52.17 |
| Mahalanobis | 42.24 | 97.36 | 41.17 | 98.24 | 47.27 | 97.32 | 56.53 | 62.78 | 46.80 | 88.93 |
| Energy | 92.66 | 39.73 | 87.40 | 52.03 | 85.17 | 57.85 | 85.42 | 52.11 | 87.66 | 50.43 |
| GradNorm | 93.40 | 26.72 | 88.81 | 40.94 | 84.09 | 52.00 | 87.68 | 43.37 | 88.49 | 40.76 |
| ReAct | **95.41** | **23.97** | 91.01 | 42.04 | 87.85 | **49.84** | 91.84 | 40.62 | 91.53 | 39.12 |
| FS+TA | 93.25 | 32.86 | **93.75** | **33.54** | **88.43** | 50.62 | **93.94** | **26.45** | **92.34** | **35.87** |

Table 14: Evaluation on near OOD benchmark.

| ID Dataset | OOD Dataset | Method | AUROC↑ | FPR95↓ |
|---|---|---|---|---|
| CIFAR10 | CIFAR100 | MSP | 88.73 | 60.48 |
| | | ODIN | 89.00 | 51.30 |
| | | Energy | 88.99 | 51.62 |
| | | GradNorm | 77.72 | 70.00 |
| | | ReAct+MSP | 88.54 | 60.51 |
| | | ReAct+ODIN | 88.76 | 50.83 |
| | | ReAct+Energy | 88.76 | 51.43 |
| | | DICE+MSP | 88.79 | 57.46 |
| | | DICE+ODIN | 88.49 | 52.79 |
| | | DICE+Energy | 86.48 | 61.30 |
| | | FS+TA+MSP | 86.81 | 57.39 |
| | | FS+TA+ODIN | 89.61 | 50.82 |
| | | FS+TA+Energy | **90.85** | **47.79** |
| CIFAR100 | CIFAR10 | MSP | 75.98 | 84.21 |
| | | ODIN | 76.80 | 83.94 |
| | | Energy | 76.68 | 84.40 |
| | | GradNorm | 67.14 | 85.59 |
| | | ReAct+MSP | 76.12 | 84.50 |
| | | ReAct+ODIN | 76.12 | 84.50 |
| | | ReAct+Energy | 76.35 | 85.00 |
| | | DICE+MSP | 75.42 | 84.53 |
| | | DICE+ODIN | 75.36 | 84.89 |
| | | DICE+Energy | 72.87 | 88.06 |
| | | FS+TA+MSP | 72.52 | **82.72** |
| | | FS+TA+ODIN | **77.49** | 82.87 |
| | | FS+TA+Energy | 77.45 | 82.99 |
| ImageNet-1k | SSB-hard | Energy | 72.34 | 83.88 |
| | | MSP | 72.16 | 84.53 |
| | | ODIN | 72.75 | 83.75 |
| | | ASH | 74.07 | 80.65 |
| | | FS+TA+Energy | **74.26** | **79.91** |
| | | FS+TA+MSP | 73.31 | 81.35 |

## B.10 EXPLORING MORE DISTANCE METRICS FOR THE FOREGROUND AND BACKGROUND COMPONENTS

We try some distance metrics shown in Table 15 and the results show that mean absolute deviation is still a simple but effective choice.

Moreover, we also try to design Feature Sim scores w.r.t. the mean value, standard deviation and mean absolute deviation of the gram matrix. The experimental results (Feature Sim with gram matrix + Threshold Activation + Energy) are shown in Table 16. Note that the gram matrix is calculated by matrix multiplication of flattened and normalized C5 features $(N, HW, C) \cdot (N, C, HW) -> (N, HW, HW)$.

It is interesting that the mean value of gram matrix does not work at all, while the standard deviation and mean absolute deviation of gram matrix work just fine. We believe this is because when the difference in foreground and background activation is significant (ID sample), the gram matrix will also show a large deviation, and vice versa. This phenomenon also confirms our foreground-background hypothesis.

For the mean of the gram matrix, since the cosine similarity between features changes between $[-1, 1]$, the meaning of feature activation strength has been lost. Therefore, directly measuring the mean of the gram matrix may have no physical meaning.

Table 15: Ablation on distance metrics for the foreground and background components.

| Statistics | iNaturalist | | SUN | | Places | | Textures | | Average | |
|---|---|---|---|---|---|---|---|---|---|---|
| | AUROC↑ | FPR95↓ | AUROC↑ | FPR95↓ | AUROC↑ | FPR95↓ | AUROC↑ | FPR95↓ | AUROC↑ | FPR95↓ |
| Mean absolute deviation (baseline) | **96.86** | **16.03** | 93.58 | 29.26 | 88.68 | 43.40 | 95.54 | 20.48 | **93.67** | **27.29** |
| Median absolute deviation | 94.90 | 24.51 | 93.75 | 28.52 | **90.71** | **38.38** | 87.17 | 52.78 | 91.63 | 36.05 |
| Standard deviation | 96.70 | 16.77 | 93.10 | 30.38 | 87.81 | 45.14 | **95.67** | **19.89** | 93.32 | 28.05 |
| Channel mean absolute deviation | 95.29 | 25.35 | **94.72** | **25.77** | 90.52 | 38.50 | 86.01 | 45.82 | 91.63 | 33.86 |

*Channel mean absolute deviation: using the mean value in channel dimension instead of spatial dimension to calculate mean absolute deviation.

Table 16: Exploring the design of FS based on the Gram Matrix.

| Statistics | iNaturalist | | SUN | | Places | | Textures | | Average | |
|---|---|---|---|---|---|---|---|---|---|---|
| | AUROC | FPR95 | AUROC | FPR95 | AUROC | FPR95 | AUROC | FPR95 | AUROC | FPR95 |
| Mean | 25.67% | 99.91% | 55.15% | 98.12% | 58.55% | 97.86% | 20.93% | 99.77% | 40.07% | 98.91% |
| Standard deviation | 96.23% | 19.46% | 88.91% | 50.77% | 86.30% | 60.63% | 96.22% | 14.59% | 91.92% | 36.36% |
| Mean absolute deviation | 95.79% | 22.07% | 88.01% | 54.59% | 85.47% | 63.54% | 95.55% | 17.73% | 91.21% | 39.48% |

## B.11 GMM-BASED SCORE

We try to use GMM-estimated foreground-background distances as OOD scores and the results are shown in Table 17. In detail, we design a GMM-based score $S_{GMM} = (\mu_f - \mu_g) \cdot \frac{w_f}{w_g}$, which takes channel-wised foreground-background statistics $\mu_f, \mu_g$ and is scaled by the proportion of the two components. The scale factor is used because a sample with a higher foreground weight is more likely to be the ID. The performance on the ImageNet benchmark is 90.83% AUROC and 35.08 % FPR95 and is close to our proposed method, illustrating that our theory explains our method properly. We think the performance gap is mainly caused by the estimation errors of GMM.

Table 17: GMM-based score.

| Method | iNaturalist | | SUN | | Places | | Textures | | Average | |
|---|---|---|---|---|---|---|---|---|---|---|
| | AUROC↑ | FPR95↓ | AUROC↑ | FPR95↓ | AUROC↑ | FPR95↓ | AUROC↑ | FPR95↓ | AUROC↑ | FPR95↓ |
| $S_{GMM}$ | 95.13 | 27.40 | 94.71 | 24.50 | 91.20 | 36.40 | 82.29 | 52.00 | 90.83 | 35.08 |

## B.12 DISCUSSION ON VIT-BASED MODELS

We have attempted to calculate the Feature Sim score directly on the token-based features of ViT. Specifically, we use the encoder features with the shape of $(B, C, L)$ as input to the OOD detector and obtain results of 75.27% AUROC and 84.69% FPR95 on the ImageNet benchmark. These results hint that the proposed Feature Sim score has the potential to distinguish OOD samples by analyzing token-based features. However, the performance was lower than that of CNNs, mainly due to the global attention mechanism in ViTs, which blurs the boundary between foreground and background and causes performance degradation. Decoupling foreground and background features may be a possible improvement in this regard.

### B.13 DISCUSSION ABOUT THE OBJECT SCALE.

We explore the impact of the foreground size in the OOD image to our method FS+TA. Among the four OOD datasets, only iNaturalist is object-centric, so it is most susceptible to the impact of object scale. Therefore, we use iNaturalist as an example for our experiments. We zoom in the OOD image w.r.t. the image center (w/2, h/2) and then resize it to 224. This way, we will get foregrounds of different sizes. We perform continuous zoom-in operations on the iNaturalist dataset, without altering the ID dataset. The experimental results of FS+TA are shown in Table 18.

Table 18: The impact of the foreground sizes in the OOD images.

| Scale Ratio | iNaturalist | |
|---|---|---|
| | AUROC | FPR95 |
| 1 | 96.86 | 16.03 |
| 1.1 | 97.84 | 10.90 |
| 1.2 | 97.92 | 10.57 |
| 1.3 | 98.08 | 9.78 |
| 1.4 | 98.17 | 9.18 |
| 1.5 | 98.25 | 8.62 |
| 1.6 | 98.30 | 8.25 |
| 1.7 | 98.33 | 8.18 |
| 1.8 | 98.34 | 8.33 |
| 1.9 | 98.34 | 8.31 |
| 2 | 98.48 | 7.66 |

The experimental results show that when the foreground size of OOD images increases, the performance of OOD detection improves. Our detailed analysis of this phenomenon is as follows:

- During the continuous enlargement of the foreground size, a mismatch occurs between the size of the object and the receptive field of the network, leading to a further reduction in the network's response to the foreground. This causes the OOD scores of iNaturalist to decrease, resulting in an increase in the OOD detection performance.
- The increase in the proportion of the foreground in OOD images does not enhance the activation intensity of the foreground. The experimental phenomenon is consistent with our previous explanation.

Moreover, we also examine some samples shown in Fig. 6, and their OOD scores also decrease as the size of the foreground increased.

## C VISUALIZATIONS

### C.1 MORE VISUALIZATIONS OF HEATMAP

Following Peng et al. (2022), we visualize output feature maps of the backbone by heatmaps in Fig. 7. Note that we derive the heatmaps by summing the features of the last convolutional layer across the channel dimension and normalizing it to [0,1].

Comparing the first and third columns, we have carefully selected challenging examples, i.e., ID and OOD foreground activation of comparable strength, to illustrate the role of Threshold Activation. These OOD data tend to make the model overconfident because of the high foreground activation. However, looking at the second and last columns, ID features retain more significant activation than OOD features after TA. TA helps to select the feature patterns that really belong to ID data.

The image names of Fig. 7 (in left-to-right, top-to-bottom order):

```
ILSVRC2012_val_00049692, 0fd78e4a67024dc68d0266b46f8fc8da (iNatu-
ralist), ILSVRC2012_val_00011802, 3e52bbb9139d5fae639880dada9405bc
(iNaturalist), ILSVRC2012_val_00020514, braided_0009
```

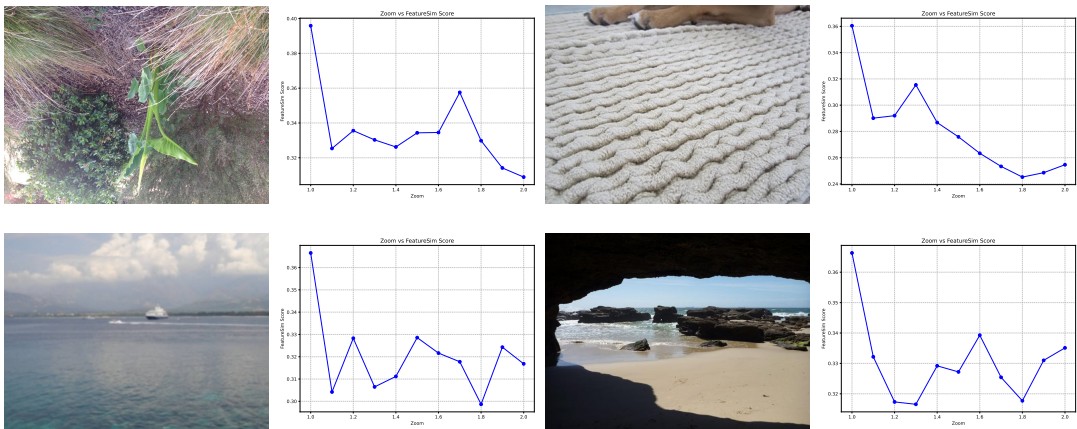

Figure 6: The change trending of FS w.r.t. foreground size. We select one image from each OOD dataset as examples.

```
(Textures), ILSVRC2012_val_00020967, g_grotto_00004308
(Places), ILSVRC2012_val_00022333, sun_aoslhdrylyvadajn (SUN)
ILSVRC2012_val_00028224, 97e0f911165a96d93f5eb82cbdf35103 (iNat-
uralist).
```

## C.2  MORE VISUALIZATIONS OF FEATURE MAP INTENSITY DISTRIBUTION

We visualize pixel-wise feature map intensity distribution of ID and OOD data in Figs. 8 and 9. Note that the feature maps are flattened in the channel, width, and height dimensions, and we count the frequency histograms of the activation intensities as probability densities. We model the feature activations with GMM and use the EM algorithm to fit them. It can be seen that the estimated distribution can fit the original data well and obtain foreground and background components. Note that we resize the original pictures for a better view.

Compared with the OOD data, the foreground of the ID data can be clearly distinguished from the background, and the ID distributions show a bimodal character. The phenomenon illustrates the effectiveness of our approach.

Looking at the last row of Fig. 9, the mean feature intensities of the two images are around 0.4, which are significantly higher than the other images in Fig. 9. These images may make the model overconfident due to their own high activation. Our method observes the distance relationship between the foreground and background components, so it can effectively avoid this problem and thus improve OOD detection performance.

The image names of Fig. 8 (in left-to-right, top-to-bottom order):

```
ILSVRC2012_val_00006757, ILSVRC2012_val_00007524,
ILSVRC2012_val_00016410, ILSVRC2012_val_00044899,
ILSVRC2012_val_00017260, ILSVRC2012_val_00020341,
ILSVRC2012_val_00022515, ILSVRC2012_val_00023116,
ILSVRC2012_val_00032032, ILSVRC2012_val_00035988,
ILSVRC2012_val_00039609, ILSVRC2012_val_00045728.
```

The image names of Fig. 9 (in left-to-right, top-to-bottom order):

```
00e019b42a68d28c224ba3abe703a60f (iNaturalist),
b0c07bdd99a728c4f6eccc61f7b4a17c (iNaturalist),
banded_0047 (Textures), c_canal_urban_00000972
(Places), d_desert_vegetation_00001079 (Places),
de2bca4aa8510da96d58cf29453dbe96 (iNaturalist),
sun_aeevojrozcqhtqgc (SUN), sun_aoqjnzwgdikwayvw (SUN),
```

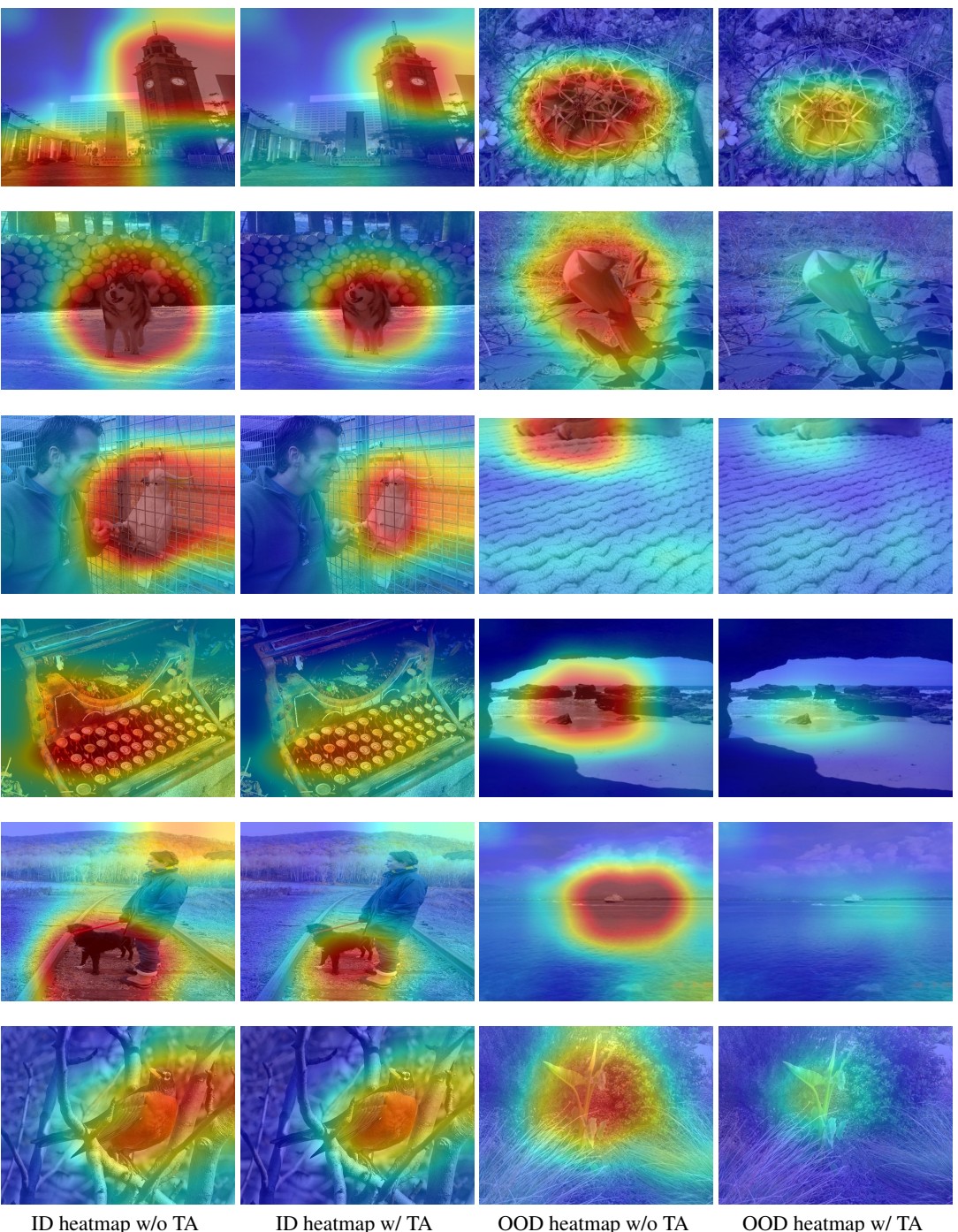

ID heatmap w/o TA      ID heatmap w/ TA      OOD heatmap w/o TA      OOD heatmap w/ TA

Figure 7: Feature visualization of ResNet50. The red part of the image represents higher feature activation. Threshold Activation (TA) is our proposed module to suppress features in the middle layer.

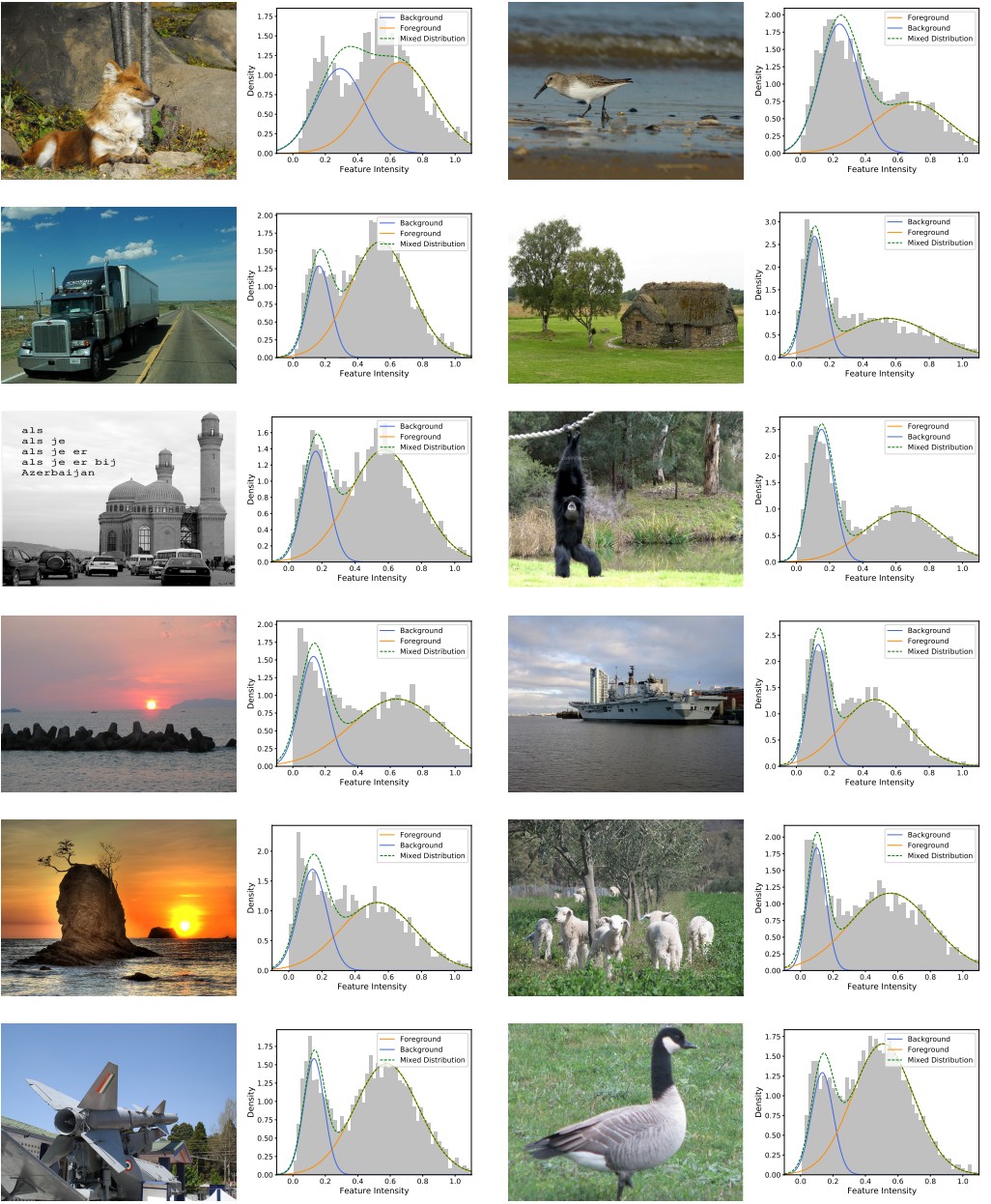

Figure 8: Pixel-wise feature map intensity distribution of ID data. The gray bar represents the histogram of real data. The blue and orange solid lines represent the EM-estimated background and foreground components of the GMM model. The green dashed line represents the mixed distribution.

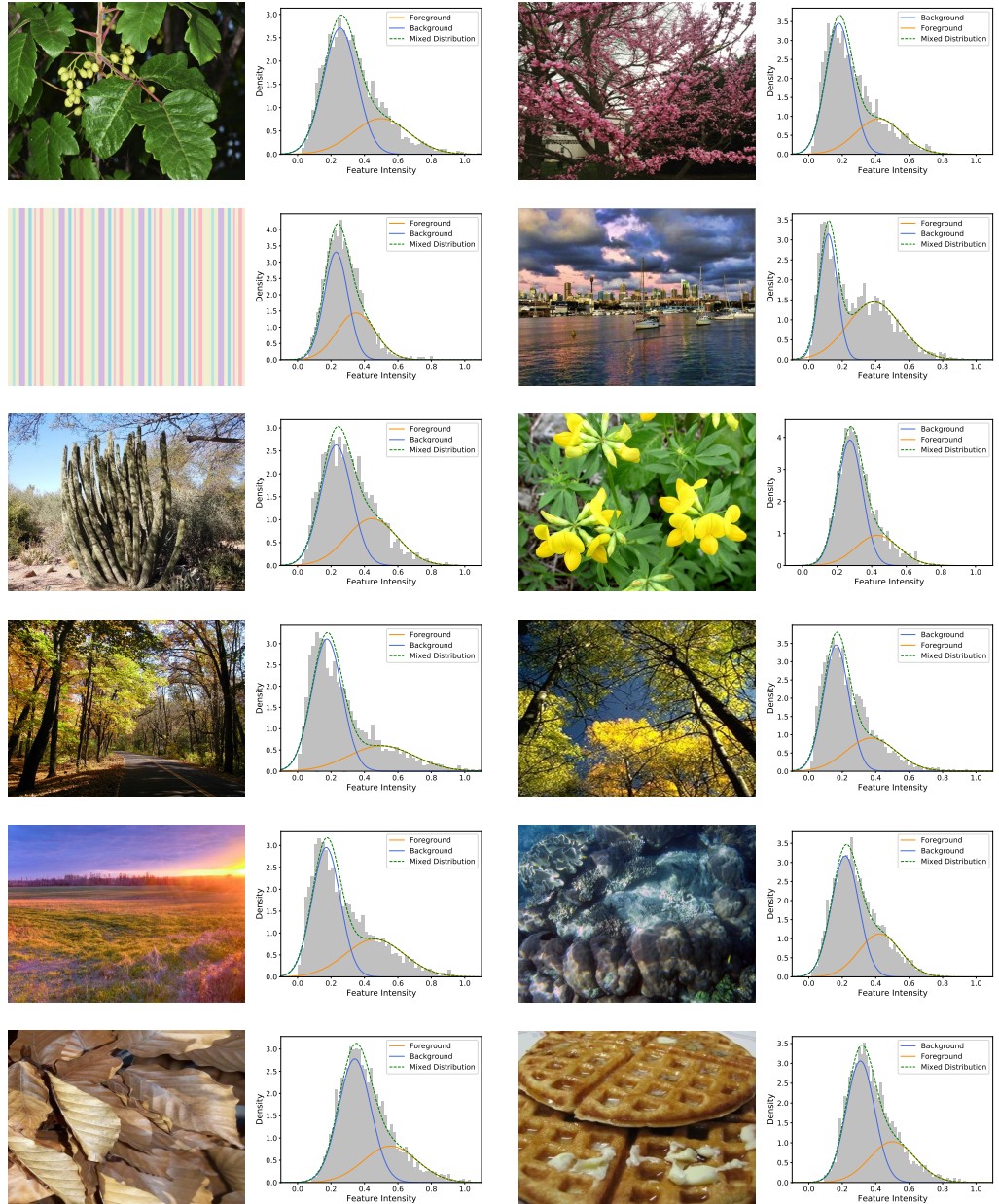

Figure 9: Pixel-wise feature map intensity distribution of OOD data. The gray bar represents the histogram of real data. The blue and orange solid lines represent the EM-estimated background and foreground components of the GMM model. The green dashed line represents the mixed distribution.

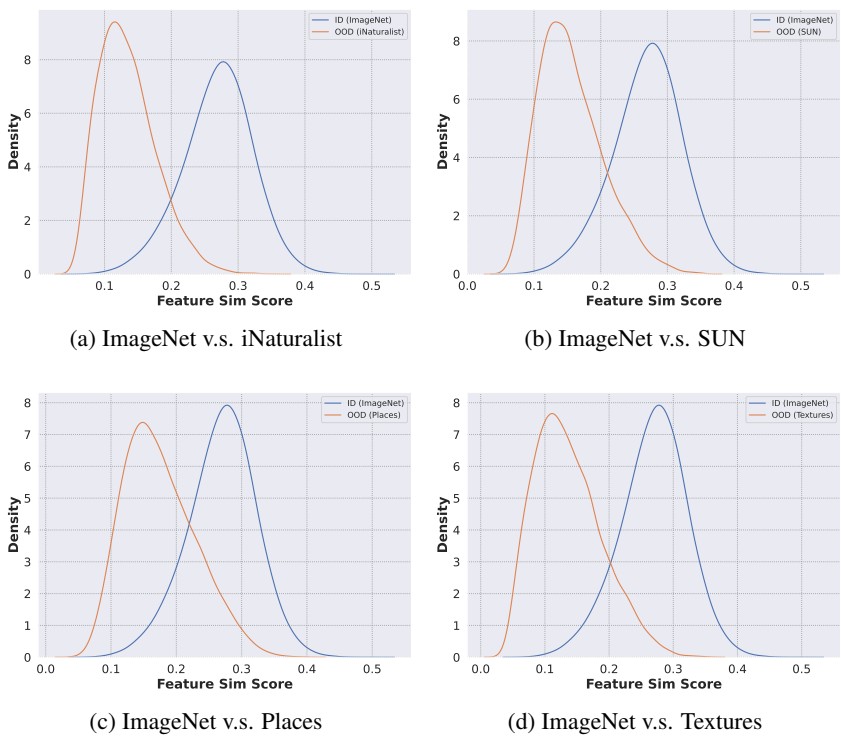

(a) ImageNet v.s. iNaturalist

(b) ImageNet v.s. SUN

(c) ImageNet v.s. Places

(d) ImageNet v.s. Textures

Figure 10: Distributions of Feature Sim score on ImageNet Benchmark.

```
t_tundra_00000203 (Places), u_underwater_ocean_deep_00002242
(Places), veined_0154 (Textures), waffled_0061 (Textures).
```

### C.3 VISUALIZATION OF FEATURE SIM SCORE DISTRIBUTION

The visualization is shown in Fig. 10 and it can be seen that the distributions of ID and OOD scores are well separated.

## D THEORETICAL DETAILS

### D.1 EXPECTATION-MAXIMIZATION (EM) ALGORITHM AND GMM

#### D.1.1 EM ALGORITHM

Here is the notation:

$$
\begin{aligned}
x_1, \ldots, x_n &\quad \text{data points} \\
z_1, \ldots, z_n &\quad \text{hidden variables, taking values } k = 1 \ldots K \\
\theta &\quad \text{parameters} \\
t &\quad \text{iteration}
\end{aligned}
$$

EM creates an iterative procedure where we update the $z_i'$'s and then update $\theta$. It is an alternating minimization scheme similar to $k$-means.

- E-step: compute cluster assignments (which are probabilistic)
- M-step: update $\theta$ (which are the clusters' properties)

Then, the log likelihood would be

$$
\begin{aligned}
\log \text{ likelihood }(\theta) &= \log P\left(X_1, \ldots, X_n = x_1, \ldots, x_n \mid \theta\right) \\
&= \sum_i \log P\left(X_i = x_i \mid \theta\right) \text{ (by independence)} \\
&= \sum_i \log \sum_k P\left(X_i = x_i, Z_i = k \mid \theta\right) \text{ (hidden variables)} \\
&= \sum_i \log \sum_k P\left(Z_i = k \mid x_i, \theta_t\right) \frac{P\left(X_i = x_i, Z_i = k \mid \theta\right)}{P\left(Z_i = k \mid x_i, \theta_t\right)}
\end{aligned}
\tag{13}
$$

The idea of EM is to find a lower bound on likelihood $(\theta)$ that involves $P(x, z \mid \theta)$. Maximizing the lower bound always leads to higher values of likelihood $(\theta)$.

According to Jensen's inequality, we have:

$$
\log \text{ likelihood }(\theta) \geq \sum_i \sum_k P\left(Z_i = k \mid x_i, \theta_t\right) \log \frac{P\left(X_i = x_i, Z_i = k \mid \theta\right)}{P\left(Z_i = k \mid x_i, \theta_t\right)} =: A\left(\theta, \theta_t\right). \tag{14}
$$

$A\left(\cdot, \theta_t\right)$ is called the auxiliary function.

Then, the steps of EM would be:

- E-step: compute $P\left(Z_i = k \mid x_i, \theta_t\right) =: \gamma_{ik}$ for each $i, k$.
- M-step: $\max_\theta A\left(\theta, \theta_t\right) = \sum_i \sum_j \gamma_{ik} \log \frac{P(X_i = x_i, Z_i = k \mid \theta)}{\gamma_{ik}}$.

### D.1.2  GMM

we can apply EM to GMM. Here is the notation:

$$
\begin{aligned}
w_{kt} &= \text{ probability to belong to cluster } k \text{ at iteration } t \\
\boldsymbol{\mu}_{kt} &= \text{ mean of cluster } k \text{ at iteration } t \\
\boldsymbol{\sigma}_{kt} &= \text{ covariance of } k \text{ at iteration } t
\end{aligned}
$$

and $\theta_t$ is the collection of $\left(w_{kt}, \boldsymbol{\mu}_{kt}, \boldsymbol{\sigma}_{kt}\right)$ 's at iteration $t$.

The likelihood of GMM is

$$
\begin{aligned}
\text{likelihood }(\theta) &= \prod_i P\left(X_i = x_i \mid \theta\right), \\
&= \prod_i \sum_{k=1}^K P\left(X_i = x_i \mid z_i = k, \theta\right) P\left(z_i = k \mid \theta\right) \quad \text{(law of total probability)} \\
&= \prod_i \sum_{k=1}^K N\left(x_i; \boldsymbol{\mu}_k, \boldsymbol{\sigma}_k\right) w_k.
\end{aligned}
\tag{15}
$$

Then, taking the log,

$$
\begin{aligned}
\log \text{ likelihood }(\theta) &= \log \prod_i \sum_k P\left(X_i = x_i \mid z_i = k, \theta\right) P\left(z_i = k \mid \theta\right) \\
&= \sum_i \log \sum_k P\left(X_i = x_i \mid z_i = k, \theta\right) P\left(z_i = k \mid \theta\right)
\end{aligned}
\tag{16}
$$

Thus, the EM for GMM is:

- E-step:

$$
P\left(Z_i = k \mid x_i, \theta_t\right) = \frac{N\left(x_i; \boldsymbol{\mu}_{kt}, \boldsymbol{\sigma}_{kt}\right) w_{kt}}{\sum_{k'} N\left(x_i; \boldsymbol{\mu}_{k't}, \boldsymbol{\sigma}_{k't}\right) w_{k't}} =: \gamma_{ik} \tag{17}
$$

- M-step: After complex calculations, the results are

$$\mu_{k,t+1} = \frac{\sum_i x_i \gamma_{ik}}{\sum_i \gamma_{ik}}$$

$$\sigma_{k,t+1} = \frac{\sum_i \gamma_{ik} (x_i - \boldsymbol{\mu}_{k,t+1}) (x_i - \boldsymbol{\mu}_{k,t+1})^T}{\sum_i \gamma_{ik}} \tag{18}$$

$$w_{k,t+1} = \frac{\sum_i \gamma_{ik}}{n}$$

where $n$ is the total sample number.

### D.2 PROOF OF EQ. (8)

*Proof.* According to Eq. (18), for any iteration $t$, we have:

$$
\begin{aligned}
w\mu_1 + (1-w)\mu_2 &= \frac{\sum_i \gamma_{i1}}{n} \frac{\sum_i x_i \gamma_{i1}}{\sum_i \gamma_{i1}} + \frac{\sum_i \gamma_{i2}}{n} \frac{\sum_i x_i \gamma_{i2}}{\sum_i \gamma_{i2}} \\
&= \frac{\sum_i x_i (\gamma_{i1} + \gamma_{i2})}{n} \\
&= \frac{\sum_i x_i}{n} \\
&= \mu
\end{aligned}
\tag{19}
$$

$\square$

### D.3 PROOF OF EQ. (10)

Assume that $Z$ has a standard normal distribution and has a probability density function $\phi$ and a distribution function $\Phi$, given by

$$\phi(z) = \frac{1}{\sqrt{2\pi}} e^{-z^2/2}, \quad z \in \mathbb{R}.$$

$$\Phi(z) = \int_{-\infty}^{z} \phi(x) dx = \int_{-\infty}^{z} \frac{1}{\sqrt{2\pi}} e^{-x^2/2} dx, \quad z \in \mathbb{R}. \tag{20}$$

*Proof.* According to Eq. (20), Eq. (7) can be written as:

$$p(\hat{h}) = w\phi((\hat{h} - \mu_1)/\sigma_1)/\sigma_1 + (1-w)\phi((\hat{h} - \mu_2)/\sigma_2)/\sigma_2. \tag{21}$$

Assume that $y = \hat{h} - \mu$, Eq. (21) can be written as:

$$p(y) = w\phi((y + \mu - \mu_1)/\sigma_1)/\sigma_1 + (1-w)\phi((y + \mu - \mu_2)/\sigma_2)/\sigma_2. \tag{22}$$

Assume that $q = |y|$, Eq. (22) can be written as:

$$
\begin{aligned}
p(q) = & w\phi((q + \mu - \mu_1)/\sigma_1)/\sigma_1 + (1-w)\phi((q + \mu - \mu_2)/\sigma_2)/\sigma_2 \\
& + w\phi((q - \mu + \mu_1)/\sigma_1)/\sigma_1 + (1-w)\phi((q - \mu + \mu_2)/\sigma_2)/\sigma_2, \quad q > 0.
\end{aligned}
\tag{23}
$$

The expectation of $q$ is :

$$
\begin{aligned}
\mathbb{E}(q) &= \int_0^{+\infty} q p(q) dq \\
&= 2w(\mu - \mu_1)\Phi(-(\mu_1 - \mu)/\sigma_1) + 2(1-w)(\mu - \mu_2)\Phi(-(\mu_2 - \mu)/\sigma_2) \\
&\quad + 2w\sigma_1\phi((\mu_1 - \mu)/\sigma_1) + 2(1-w)\sigma_2\phi((\mu_2 - \mu)/\sigma_2). \\
&= w\mathbb{E}(f^*) + (1-w)\mathbb{E}(g^*).
\end{aligned}
\tag{24}
$$

$\square$

# E  COMPARISON WITH RELATED WORKS

## E.1  DIFFERENCE BETWEEN OUR METHOD AND REACT

React believes that abnormally high activations in the fully connected layer can lead to classifier overconfidence, so it truncates all activations that are higher than the threshold. On the other hand, our Threshold Activation (TA) method starts from the phenomenon of activations in feature maps and truncates all activations in intermediate layers that are lower than the threshold while weakening the other activations. This approach weakens a large number of OOD features that could potentially generate abnormal activations, preventing the foreground parts of OOD images from producing excessively high activations. While React and TA seem to use a threshold to handle feature activations, their operational methods are entirely different.

The Threshold Activation (TA) and ReAct methods handle features at different positions in the network, leading to opposite choices in retaining features. TA has sufficient network parameters following it to reprocess and restore features, allowing it to choose to retain salient (high) activations. On the other hand, ReAct has only one fully connected layer after it, limiting the network's ability to further process features. As a result, ReAct chooses to retain stable (low) activations. The differences in the types of features they handle (spatial features vs. classification features) and their positions in the network (intermediate convolutional layers vs. classifier layer) result in the adoption of completely different feature selection approaches between TA and ReAct.

In addition, as shown in Table 2, using TA and ReAct together achieves better results. This indicates that TA and ReAct are complementary methods. TA's processing of convolutional features and ReAct's handling of fully connected classification features bring different gains to OOD detection. Experimental results demonstrate that these two gains can be combined, showing that the benefits of using TA and ReAct are cumulative.

In summary, we believe that Threshold Activation and ReAct have fundamental difference.

## E.2  DIFFERENCE BETWEEN OUR METHOD AND ASH

ASH is an plug-in algorithm that requires existing score functions to work in conjunction with it, such as MSP and Energy. Furthermore, while ASH claims to operate on feature map activations, it actually works well when placed in the penultimate layer of the classifier, so it does not effectively process feature map activations as claimed.

Essentially, we believe that ASH truncates features, whereas our proposed TA weakens them.

We have noticed that the core idea behind the three forms of ASH is to set the activations below the threshold to zero and preserve or even amplify those above the threshold. However, due to the characteristics of convolutional kernels, the remaining high OOD responses can "diffuse" back to other regions during the forward propagation of the network, recontaminating the "purified" feature maps (see Fig. 11). This also explains why placing the ASH method in non-classifier positions leads to poor results (see Table 19).

In contrast, TA has some similarities in design with ASH but carries a different essence. Firstly, we observe that the absolute strength of responses for ID features is higher than that of OOD features in the feature maps. Therefore, if we weaken both ID and OOD features by the same threshold, the information preserved in ID features will be much more than that in OOD features (see Fig. 11). Secondly, this weakening effect avoids the "contamination" issue seen in ASH's feature maps, ensuring that OOD, after being weakened, will not generate high activations to mislead classifier's outputs.

## E.3  DIFFERENCE BETWEEN OUR METHOD AND MAHALANOBIS

Mahalanobis precomputes the mean and normalize the feature amplitude of each class using the covariance of all training samples. During the inference stage, they calculate the Mahalanobis distance difference between the input sample's feature and the class prior feature.

On the other hand, the Feature Sim method examines the difference between the sample's feature map and its mean, reflecting the significance of feature activation differences between ID and OOD

Table 19: OOD detection performance comparison on the ImageNet-1k benchmark. All methods are based on the same model ResNet50 trained on ID data only. All values are percentages. ASH* means that we only use ASH-P for feature truncation after the C4 stage of ResNet50.

| Method | iNaturalist | | SUN | | Places | | Textures | | Average | |
|---|---|---|---|---|---|---|---|---|---|---|
| | AUROC↑ | FPR95↓ | AUROC↑ | FPR95↓ | AUROC↑ | FPR95↓ | AUROC↑ | FPR95↓ | AUROC↑ | FPR95↓ |
| FS+TA | 96.86 | 16.03 | 93.58 | 29.26 | 88.68 | 43.40 | 95.54 | 20.48 | 93.67 | 27.29 |
| FS+ASH* | 91.78 | 34.29 | 83.80 | 54.21 | 77.13 | 66.83 | 95.27 | 19.47 | 86.99 | 43.70 |

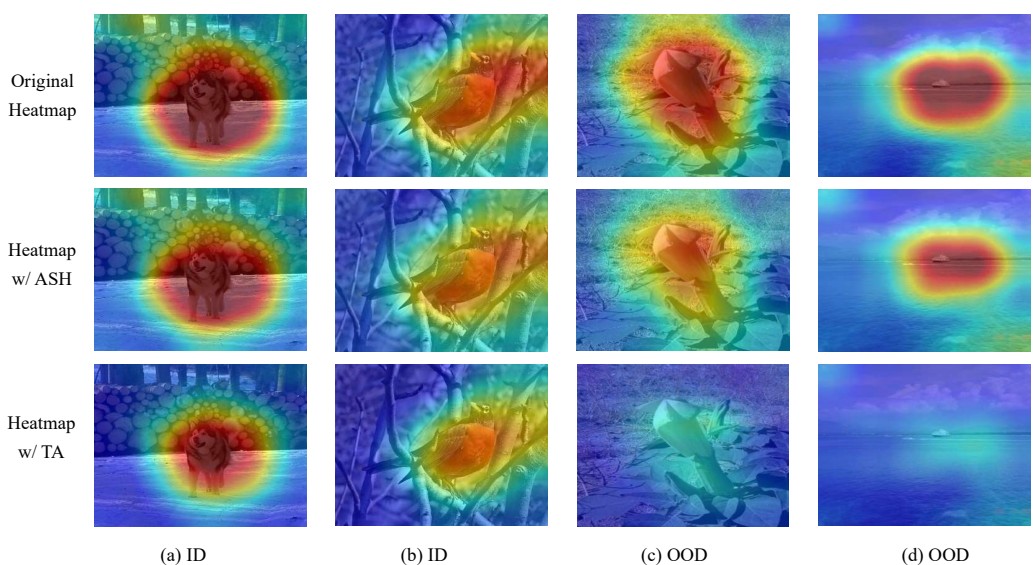

(a) ID      (b) ID      (c) OOD      (d) OOD

Figure 11: Activation visualization of ID and OOD samples. The original heatmaps (Row 1) are obtained from the C5 block of ResNet50 trained on ImageNet-1k as ID. The last two rows are heatmaps from C5 while TA/ASH are placed in C4 block. From OOD heatmaps, we notice that TA can effectively weaken OOD activations while ASH suffers from the "feature pollution" issue – the remaining OOD features are reactivated by convolutional layers.

samples. Unlike Mahalanobis-based methods, our proposed Feature Sim method does not rely on prior class distribution and does not require obtaining statistical features from training samples. Instead, it mines clues from the sample's own foreground-background relationship. We believe that these two types of methods are fundamentally different in their approach.

## F  CODES

The core functions are shown below and the detailed codes are available at https://anonymous.4open.science/r/feature_map/. The proposed methods have quite simple and elegant code implementations as follows.

```python
import torch

def threshold_activation(x, k):
    x = torch.nn.functional.relu(x - k)
    return x

def feature_sim(x):
    B, C, H, W = x.shape
```

```
x = x.reshape(B, C, H * W)
x_mean = x.mean(dim=-1).unsqueeze(-1)  # (B, C, 1) Equivalent to the
↪    GAP operator
ood_score = torch.abs(x - x_mean).mean(dim=(-1, -2))  # (B,)
return ood_score
```

