# OpenReview forum: "Feature Map Matters in Out-of-distribution Detection"
_ICLR.cc/2024/Conference — Submitted to ICLR 2024_

### Official Review · Reviewer_NfBj · 2023-10-28

**Soundness:** 3 good
**Presentation:** 3 good
**Contribution:** 3 good
**Rating:** 5
**Confidence:** 3

**Summary:**

This paper propose two methods to increase OOD performance:
(1) a score function called Feature Sim (FC), which is the mean absolute deviation of feature maps.This rely on the assumption that the activation difference between the foreground and background of the ID data is larger than OOD data.
(2) a Threshold Activation (TA) module which is a shifted ReLU applied on the feature maps, the underlying assumption is that ID features have higher activations and significant foreground component than OOD feature.
Combining FS and TA with ASH gives state-of-the-art results.

**Strengths:**

1. FS measures the mean of the channel-wise deviation of the feature map: The underlying assumption of the FS is "The activation of the network for ID data is usually concentrated in the foreground (salient area), while the activation difference between foreground and background (other areas) in OOD features is not as significant." the author models this separation using GMM and provide in-depth theoretical analysis.

2. TA module enlarge the distance between ID and OOD data by weaking the lower part of the activations.

3. FS+TA+ ASH achieves state-of-the-art result and the author also provided comprehensive experiments on various dataset.

**Weaknesses:**

(1) The GMM assumption, as illustrated in Figure 3 also C.2, relies on the fact that ID has object, while OOD data has only background. However, this is not clearly defined in the ID and OOD data definition. In fact, for near OOD task, ID and OOD data both contain objects. (e.g. ImageNet vs. SSB-hard, MINIST vs. CIFAR).

(2) The author should provide more details on the hyperparameter $\labmda$ for Section 5.3. For example, how to choose it? In the experiments, is it chosen by validating on validation dataset or testing dataset? Is there any ablation on this factor?

(3) Although the author shows state-of-the-art results combing FS and TA with ASH (+energy score), this treats FS and TA as orthogonal methods with ASH and Energy Score. However this is not true, as FS is an OOD scores and TA is a model rectification technique, the author should compare them separately and directly with their opponents.
- compare FS with other OOD score function: Energy, MLS, MSP.
- compare TA with other post-hoc model rectification technique: ReAct, DICE, ASH

**Questions:**

Regarding FS:

(1) For Eq.4, what is the GAP operator?

(2) FS measures the mean of the channel-wise deviation of the feature map, what is the intuition for this design? What is the difference if using the deviation of the feature map?

Regarding TA:

(3) The design of TA is opposite to ReAct, how this leads to the same outcome (distancing ID and OOD data) while the changes are opposite to ReAct?

(4) For the choices of the stage to apply TA and FS, are they validated directly in the testing dataset?

I am willing to raise my score if the author can address the above issues.

---

> ### Author Response · Authors · 2023-11-20
>
> >**W1:** The GMM assumption, as illustrated in Figure 3 also C.2, relies on the fact that ID has object, while OOD data has only background. However, this is not clearly defined in the ID and OOD data definition. In fact, for near OOD task, ID and OOD data both contain objects. (e.g. ImageNet vs. SSB-hard, MINIST vs. CIFAR).
>
>
> **A1:** Thanks for your comments!  We'd like to clarify a misconception that OOD data consists only of background elements. In fact, OOD images also include foreground elements, such as various plants in the iNaturalist dataset or objects in scenes (like fountains, trees, boats, etc.) in the SUN and Places datasets. **For example, the foreground of the OOD image in Figure 3 is a waterfall that is also the Ground Truth of the image.**
>
> In our paper, “foreground” does not merely refer to the image contents corresponding to its label; rather, “foreground” pertains to the regions or objects that play a critical role in determining the sample labels. The definition of “foreground” in our paper aligns with that of saliency detection [1], i.e., saliency detection aims to locate the important parts of natural images which attract our attention [2]. For example, in an indoor scene, a model doesn't label the image based on plain walls but focuses on distinctive objects like furniture, which form high-activation "foreground" regions. In contrast, less significant elements like walls are considered "background" with lower activations.
>
> Our hypothesis is based on the premise that OOD samples exhibit lower foreground activation than ID samples. As illustrated in Figure 3 and Appendix C.2, this occurs because the model, trained on ID images, is less sensitive to foreground objects associated with OOD, leading to lower foreground activations. This principle underpins the effectiveness of our method.
>
> The results for ImageNet vs. SSB-hard and MINIST vs. CIFAR are shown below. Our method still can perform well on these settings.
>
> | Method       | ID          | OOD      | AUROC      | FPR95      |
> |--------------|-------------|----------|------------|------------|
> | MSP          | CIFAR100    | MNIST    | 92.55%     | 42.68%     |
> | Energy       | CIFAR100    | MNIST    | 98.29%     | 9.37%      |
> | ODIN         | CIFAR100    | MNIST    | 97.09%     | 18.36%     |
> | ASH          | CIFAR100    | MNIST    | 92.54%     | 47.15%     |
> | FS+TA+Energy | CIFAR100    | MNIST    | **99.84%** | **0.15%**  |
> | FS+TA+MSP    | CIFAR100    | MNIST    | 98.69%     | 7.06%      |
> | MSP          | CIFAR10     | MNIST    | 92.86%     | 48.73%     |
> | Energy       | CIFAR10     | MNIST    | 96.48%     | 21.58%     |
> | ODIN         | CIFAR10     | MNIST    | 96.33%     | 22.58%     |
> | ASH          | CIFAR10     | MNIST    | 97.22%     | 22.50%     |
> | FS+TA+Energy | CIFAR10     | MNIST    | 99.33%     | 3.34%      |
> | FS+TA+MSP    | CIFAR10     | MNIST    | **99.55%** | **0.12%**  |
> | Energy       | ImageNet-1k | SSB-hard | 72.34%     | 83.88%     |
> | MSP          | ImageNet-1k | SSB-hard | 72.16%     | 84.53%     |
> | ODIN         | ImageNet-1k | SSB-hard | 72.75%     | 83.75%     |
> | ASH          | ImageNet-1k | SSB-hard | 74.07%     | 80.65%     |
> | FS+TA+Energy | ImageNet-1k | SSB-hard | **74.26%** | **79.91%** |
> | FS+TA+MSP    | ImageNet-1k | SSB-hard | 73.31%     | 81.35%     |
>
>
>
> [1] Qin, Xuebin, et al. “U2-Net: Going deeper with nested U-structure for salient object detection.” Pattern recognition 106 (2020): 107404.
> [2] Zhao, Ting, and Xiangqian Wu. “Pyramid feature attention network for saliency detection.” Proceedings of the IEEE/CVF conference on computer vision and pattern recognition. 2019.

---

> > ### Author Response · Authors · 2023-11-20
> >
> > >**W2:** The author should provide more details on the hyperparameter $\lambda$
> >  for Section 5.3. For example, how to choose it? In the experiments, is it chosen by validating on validation dataset or testing dataset? Is there any ablation on this factor?
> >
> > **A2:** Thanks for your suggestions! In response to the inquiry regarding the selection of the hyperparameter $\lambda$ in our study, we'd like to emphasize its non-sensitivity and robustness, as substantiated by Figure 4 in Appendix. Our TA+FS method and the compared methods, such as ODIN and Energy, operate on distinct dynamic ranges, resulting in substantial differences in the $\lambda$ values when combined. For example, ODIN's range is narrowly set between [0.001001, 0.001029], within a $3\sigma$ range, whereas Energy spans from [4.04, 25.46]. This disparity in ranges can lead to $\lambda$ values differing by orders of magnitude.
> >
> > The rationale behind introducing $\lambda$ is its ability to normalize these varying ranges to a similar interval. This normalization ensures that the fusion weights of the two methods are approximately balanced and is independent of any specific data distribution. It relies solely on the inherent characteristics of the methods being combined, thereby eliminating concerns about test set leakage.
> >
> > Furthermore, the determination of $\lambda$ is not the result of an exhaustive search. Instead, it involves a straightforward adjustment of the dynamic ranges of the two scores to a comparable scale. This approach ensures that the fused score function effectively integrates insights from both convolutional features and the classifier. Our approach avoids the dominance of one method over the other, and $\lambda$ has demonstrated stability and effectiveness across a broad range of values.
> >
> >
> > >**W3:** Although the author shows state-of-the-art results combing FS and TA with ASH (+energy score), this treats FS and TA as orthogonal methods with ASH and Energy Score. However this is not true, as FS is an OOD scores and TA is a model rectification technique, the author should compare them separately and directly with their opponents.
> > (1)compare FS with other OOD score function: Energy, MLS, MSP.
> > (2)compare TA with other post-hoc model rectification technique: ReAct, DICE, ASH
> >
> > **A3:** There is a misunderstaing. **We did not state that FS and TA are orthogonal in our submission.** Actually, TA is a special design for FS.
> > Our proposed FS and TA are not two independent modules, but a synergistic system. It can even be argued that Threshold Activation is designed to facilitate the performance of Feature Sim.
> >
> > Specifically, FS measures the activation differences between the foreground and background of samples. For unprocessed raw feature maps, the activation differences between ID and OOD are not significant, limiting the effectiveness of Feature Sim (as shown in Table 3, without TA, FS can only achieve only 83.32\% AUROC).
> >
> > The goal of TA is to increase the foreground-background difference for ID and reduce it for OOD. We demonstrate the behavior of TA in Fig. 10 of Appendix E.2. First, we observe that the absolute strength of responses for ID features is higher than that of OOD features in the feature maps. Therefore, if we weaken both ID and OOD features by the same value 'k', the information preserved in ID features will be much more than that in OOD features.
> >
> > Then, since TA is located in the middle layer of the network, after the features are rectified, they still need to undergo deep processing. As shown in the third row of Fig. 10, the rectified ID features almost regain their original response strength after passing through the C5 stage; whereas the rectified OOD features nearly disappear after passing through the C5 stage. This is because the network is more sensitive to ID features, knowing how to extract and process ID semantics from subtle clues, even if the features are damaged in the middle layer, the network can restore them. For OOD, the high response comes from coincidentally activated textures and semantics, and when they are weakened, the network can no longer restore their original information, because it is not accustomed to processing them.
> >
> > Based on this working mechanism, the Threshold Activation at C4 provides more significant working conditions for Feature Sim at C5, and they work together effectively. In fact, we also reported the individual effects of TA and FS in Tables 2 and 3, but we believe this should only be seen as a kind of ablation reference. In summary, we believe that FS+TA should be considered as a whole rather than separate components, and it is unfair to evaluate their performance separately.

---

> > > ### Author Response · Authors · 2023-11-20
> > >
> > > >**Q1:** For Eq.4, what is the GAP operator?
> > >
> > > **A4:** Thanks for your comments! The GAP operator represent the global average pooling layer on the spatial (height, width) axis. For each channel, a feature map with shape of (H, W) will be pooled as a single value. To be specific, the shape of f(x) is (B,C,H,W) and the shape of GAP(f(x)) is (B,C).
> > >
> > > >**Q2:** FS measures the mean of the channel-wise deviation of the feature map, what is the intuition for this design? What is the difference if using the deviation of the feature map?
> > >
> > > **A5:** Thanks for your suggestions! The use of mean absolute deviation as an OOD score is based on measuring the self-similarity of features, specifically the differences among each pixel within the feature. This approach stems from the need to quantify how distinct the foreground is from the background in an image. Since there's no absolute standard for each image's activations, only a relative one, the focus shifts to measuring the disparity between the image's own foreground and background. Various scores, including the most intuitive deviation (Standard deviation), are evaluated through ablation studies in Appendix B.10. It is found that mean absolute deviation offers the most balanced performance, while the simplest form of deviation (Standard deviation) also yields satisfactory results.
> > >
> > > Mean absolute deviation: $|x_1 - \mu| + |x_2 - \mu| + ... + |x_N - \mu|$
> > >
> > > Deviation: $(x_1 - \mu)^2 + (x_2 - \mu)^2 + ... + (x_N - \mu)^2$
> > >
> > > When extreme outliers are present, the standard deviation will be considerably larger than the mean absolute deviation.
> > >
> > > |Statistics                   | iNaturalist |        |   SUN  |        | Places |        | Textures |        |         Average    |        |
> > > |:-----------------:|:-----------:|:------:|:------:|:------:|:------:|:------:|:--------:|:------:|:------------------:|:------:|
> > > |                   |    AUROC    |  FPR95 |  AUROC |  FPR95 |  AUROC |  FPR95 |   AUROC  |  FPR95 |        AUROC       |  FPR95 |
> > > | Mean absolute deviation   (baseline) |    96.86%   | 16.03% | 93.58% | 29.26% | 88.68% | 43.40% |  95.54%  | 20.48% |       93.67%       | 27.29% |
> > > |       Median absolute deviation      |    94.90%   | 24.51% | 93.75% | 28.52% | 90.71% | 38.38% |  87.17%  | 52.78% |       91.63%       | 36.05% |
> > > |        Standard deviation         |    96.70%   | 16.77% | 93.10% | 30.38% | 87.81% | 45.14% |  95.67%  | 19.89% |       93.32%       | 28.05% |
> > > |   Channel   mean absolute deviation   |    95.29%   | 25.35% | 94.72% | 25.77% | 90.52% | 38.50% |  86.01%  | 45.82% |       91.63%       | 33.86% |
> > >
> > >  *Channel  mean absolute deviation: using the mean value in channel dimension instead of spatial dimension to calculate mean absolute deviation.
> > >
> > >
> > > >**Q3:** The design of TA is opposite to ReAct, how this leads to the same outcome (distancing ID and OOD data) while the changes are opposite to ReAct?
> > >
> > >
> > > **A6:** Thanks for your comments! React believes that *abnormally high activations* in the fully connected layer can lead to classifier overconfidence, so it truncates all activations that are **higher than** the threshold. On the other hand, our Threshold Activation (TA) method starts from the phenomenon of activations in feature maps and truncates all activations in intermediate layers that are **lower than** the threshold while weakening the other activations. This approach weakens a large number of OOD features that could potentially generate abnormal activations, preventing the foreground parts of OOD images from producing excessively high activations. While React and TA seem to use a threshold to handle feature activations, their operational methods are entirely different.
> > >
> > > The Threshold Activation (TA) and ReAct methods handle features at different positions in the network, leading to opposite choices in retaining features. TA has sufficient network parameters following it to reprocess and restore features, allowing it to choose to retain **salient (high)** activations. On the other hand, ReAct has only one fully connected layer after it, limiting the network's ability to further process features. As a result, ReAct chooses to retain **stable (low)** activations. The differences in the types of features they handle (spatial features vs. classification features) and their positions in the network (intermediate convolutional layers vs. classifier layer) result in the adoption of completely different feature selection approaches between TA and ReAct.
> > >
> > > In addition, as shown in Table 2, using TA and ReAct together achieves better results. This indicates that TA and ReAct are complementary methods. TA's processing of convolutional features and ReAct's handling of fully connected classification features bring different gains to OOD detection. Experimental results demonstrate that these two gains can be combined, showing that the benefits of using TA and ReAct are cumulative.

---

> > > > ### Author Response · Authors · 2023-11-20
> > > >
> > > > >**Q4:** For the choices of the stage to apply TA and FS, are they validated directly in the testing dataset?
> > > >
> > > > **A4:** Thanks for your comments! We think FS must be placed after the C5 stage. The ablation study in Table 3 is just to demonstrate the correctness of our foreground-background theory, showing that FS can be effective at multiple positions. However, placing FS after C5 stage truly leverages the network's full capability to process and analyze images.
> > > >
> > > >
> > > > Indeed, there are multiple choices for TA, but its performance varies significantly when placed after each stage; when it is positioned after C4, the performance of FS+TA is much higher than at other positions. We follow the approach of DICE and use Gaussian noise images as a validation set to select the position for TA.
> > > >
> > > > We will add more details in the updated paper.

---

> > > > > ### Author Response · Authors · 2023-11-22
> > > > >
> > > > > Dear reviewer,
> > > > >
> > > > > Thank you for dedicating your time and effort to offer insightful comments. We have covered all your concerns in our responses. We are looking forward to your reply.

---

> > > > > > ### Comment · Reviewer_NfBj · 2023-11-23
> > > > > >
> > > > > > Thanks for the rebuttal, which addresses some of my questions. However, I still find the definition of “foreground” and “background” unclear, and I suggest the authors to define them rigorously using “saliency”. Furthermore, FS+TA perform worse than the current SOTA ASH, although stacking them to ASH provides some improvement. I maintain our original score of 5.

---

> > > > > > > ### Author Response · Authors · 2023-11-23
> > > > > > >
> > > > > > > We appreciate your time and effort in reviewing our paper. Regarding your questions, we would like to provide further clarification:
> > > > > > >
> > > > > > > We have applied  "saliency" for the definition of foreground and background in Section 4. Additionally, we claim that without leveraging classifier information, FS+TA can achieve approximately 93% AUROC. When combined with classifier information, our method achieve the state-of-the-art. Both of these points are our contributions.

---

### Official Review · Reviewer_gmH7 · 2023-10-30

**Soundness:** 3 good
**Presentation:** 3 good
**Contribution:** 3 good
**Rating:** 6
**Confidence:** 4

**Summary:**

The authors propose one simple yet effective method Feature Sim (FS) and one plug-in module Threshold Activation (TA). Experiments demonstrate their method can achieve good performance.

**Strengths:**

This paper is well written, and the concept of "foreground" and "background” is interesting. The experiments are comprehensive, and the results are promising.

**Weaknesses:**

1)	In Figure 1 and Figure 6, the authors give the feature visualization of ResNet50 (perhaps using the last layer, as claimed in Appendix C.1). However, as far as I know, in ResNet, each layer has a multitude of feature maps. Especially in the middle layers, most feature visualizations on the feature maps can’t be directly to be utilized for visualization. The patterns FS utilizes maybe like ReAct, leveraging abnormally high activations. Could the authors provide clarification about this?
2)	Feature Sim Score essentially quantifies the data dispersion (like variance) on each feature map. It is more like utilizing extreme values on intermediate feature layers, a concept that has been extensively explored by previous feature-based methods. Could the authors clarify the difference?
3)	Threshold Activation appears to borrow the concept from ReAct and apply it to intermediate layers. Could the authors clarify the novelty of TA?

**Questions:**

Shown in Weaknesses.

---

> ### Author Response · Authors · 2023-11-20
>
> >**W1:** In Figure 1 and Figure 6, the authors give the feature visualization of ResNet50 (perhaps using the last layer, as claimed in Appendix C.1). However, as far as I know, in ResNet, each layer has a multitude of feature maps. Especially in the middle layers, most feature visualizations on the feature maps can’t be directly to be utilized for visualization. The patterns FS utilizes maybe like ReAct, leveraging abnormally high activations. Could the authors provide clarification about this?
>
>
> **A1:** Sorry for the misunderstanding. We follow [1] to conduct the visualizations. To be specific, we derive the heatmap by summing the features of last convolutional layer across the channel dimension and normalizing it to [0, 1]. Concretely, assume we get a feature map after C5 block of ResNet50 with shape (1,C,H,W), note that batchsize is equal to 1. To get the visualization result, the feature map is summed along the channel dimension to be (1,H,W), and then each pixel is normalized to [0,1] to get the heatmap.
>
> ReAct operates under the premise that OOD samples will exhibit abnormally high activations, leading to the overconfidence on OOD samples. However, Feature Sim is based on the opposite concept, utilizing the principle that ID samples will have higher activations as models are more farmilar with ID samples. More detailed analysis is shown in the following answer **A3**.
>
> [1] Xiangyu Peng, Kai Wang, Zheng Zhu, Mang Wang, and Yang You. Crafting better contrastive views for siamese representation learning. In CVPR, 2022.
>
> >**W2:** Feature Sim Score essentially quantifies the data dispersion (like variance) on each feature map. It is more like utilizing extreme values on intermediate feature layers, a concept that has been extensively explored by previous feature-based methods. Could the authors clarify the difference?
>
> **A2:**  Thanks for your suggestions! We are not entirely certain which specific previous feature-based methods are referred to in the question. Therefore, we would like to focus on comparing and explaining in relation to ASH and Mahalanobis methods (already stated in Appendix E).
>
> **Difference between our method and ASH:**
>
> ASH is an plug-in algorithm that requires existing score functions to work in conjunction with it, such as MSP and Energy.  Furthermore, while ASH claims to operate on feature map activations, it actually works well when placed in the penultimate layer of the classifier, so it does not effectively process feature map activations as claimed.
>
> Essentially, we believe that ASH truncates features, whereas our proposed TA weakens them.
>
> We have noticed that the core idea behind the three forms of ASH is to set the activations below the threshold to zero and preserve or even amplify those above the threshold. However, due to the characteristics of convolutional kernels, the remaining high OOD responses can "diffuse" back to other regions during the forward propagation of the network, recontaminating the "purified" feature maps (see Fig. 10 and Table 16 in the Appendix). This also explains why placing the ASH method in non-classifier positions leads to poor results.
>
> In contrast, TA has some similarities in design with ASH but carries a different essence. Firstly, we observe that the absolute strength of responses for ID features is higher than that of OOD features in the feature maps. Therefore, if we weaken both ID and OOD features by the same value 'k', the information preserved in ID features will be much more than that in OOD features (see Fig. 10 and Table 16 in the Appendix). Secondly, this weakening effect avoids the "contamination" issue seen in ASH's feature maps, ensuring that OOD, after being weakened, will not generate high activations to mislead classifier's outputs.
>
> **Difference between our method and Mahalanobis:**
>
>
> The Mahalanobis-based methods precompute the mean and normalize the feature amplitude of each class using the covariance of all training samples. During the inference stage, they calculate the Mahalanobis distance difference between the input sample's feature and the class prior feature.
>
> On the other hand, the Feature Sim method examines the difference between the sample's feature map and its mean, reflecting the significance of feature activation differences between ID and OOD samples. Unlike Mahalanobis-based methods, our proposed Feature Sim method does not rely on prior class distribution and does not require obtaining statistical features from training samples. Instead, it mines clues from the sample's own foreground-background relationship. We believe that these two types of methods are fundamentally different in their approach.

---

> > ### Author Response · Authors · 2023-11-20
> >
> > >**W3:** Threshold Activation appears to borrow the concept from ReAct and apply it to intermediate layers. Could the authors clarify the novelty of TA?
> >
> > **A3:** Thanks for your suggestions! Here is the comparison between our method and ReAct:
> >
> > React believes that *abnormally high activations* in the *fully connected layer* can lead to classifier overconfidence, so it truncates all activations that are **higher than** the threshold. On the other hand, our Threshold Activation (TA) method starts from the phenomenon of activations in feature maps and truncates all activations in intermediate layers that are **lower than** the threshold while weakening the other activations. This approach weakens a large number of OOD features that could potentially generate abnormal activations, preventing the foreground parts of OOD images from producing excessively high activations. While React and TA seem to use a threshold to handle feature activations, their operational methods are entirely different.
> >
> > The Threshold Activation (TA) and ReAct methods handle features at different positions in the network, leading to opposite choices in retaining features. TA has sufficient network parameters following it to reprocess and restore features, allowing it to choose to retain **salient (high)** activations. On the other hand, ReAct has only one fully connected layer after it, limiting the network's ability to further process features. As a result, ReAct chooses to retain **stable (low)** activations. The differences in the types of features they handle (spatial features vs. classification features) and their positions in the network (intermediate convolutional layers vs. classifier layer) result in the adoption of completely different feature selection approaches between TA and ReAct.
> >
> > In addition, as shown in Table 2, using TA and ReAct together achieves better results. This indicates that TA and ReAct are complementary methods. TA's processing of convolutional features and ReAct's handling of fully connected classification features bring different gains to OOD detection. Experimental results demonstrate that these two gains can be combined, showing that the benefits of using TA and ReAct are cumulative.
> >
> > Based on the above content, we believe that Threshold Activation and ReAct have fundamental differences and TA should not be considered as incremental work of ReAct.

---

### Official Review · Reviewer_sGRf · 2023-10-31

**Soundness:** 3 good
**Presentation:** 3 good
**Contribution:** 2 fair
**Rating:** 6
**Confidence:** 4

**Summary:**

This paper proposes a new perspective to utilize the feature maps for OOD detection. A Feature Similarity score function, which is defined as the absolute difference between the feature map and its mean value, is proposed to distinguish the ID and OOD data. Besides, a Threshold Activation module is introduced to increase the feature separability between ID and OOD data. The performance of the proposal looks good.

**Strengths:**

1. The motivation is clear and the whole method makes sense.
2.  The method is simple and extensive experiments are performed to demonstrate the effectiveness of the proposal.
3.  The method can collaborate well with previous post-hoc methods, and FS+TA+ASH can achieve state-of-the-art on various OOD detection benchmarks.

**Weaknesses:**

1. After Eq.3 “Feature Sim measures the self-similarity of the feature maps, whose core idea is to compare the activation differences between foreground and background on the ID and OOD feature maps”, It is a little bit confused why Feature Sim compares the activation differences between foreground and background on the ID and OOD feature. More explanations should be provided.
2. "Eq. (2) is equivalent to the GAP operator". Actually, Eq. 2 is not a GAP operator, GAP performs global pooling in the feature map, whereas Eq. 2 performs pooling across different channels.
3. In section A.1, the authors train the ResNet50 from scratch, while other comparison methods (Energy, ReAct, DICE ect.) use the off-the-shelf ResNet model for experiments. Therefore, the comparisons might be unfair, I recommend the author to utilize the off-the-shelf model.
4. The Threshold Activation module is quite similar to ReAct and ASH, which makes the novelty of the introduced TA module limited. Besides, the TA module decreases the ID classification accuracy significantly, even it can be remedied by using another classification branch.

**Questions:**

1. It is not clear why employ the absolute difference between the feature map and its mean value as OOD score. Have the authors considered to use the mean value of Gram Matrix as the similarity score, which I think should be better than the absolute difference. The Gram Matrix captures the similarity between different feature maps [1].

[1] Image Style Transfer Using Convolutional Neural Networks.

---

> ### Author Response · Authors · 2023-11-20
>
> >**W1:** After Eq.3 “Feature Sim measures the self-similarity of the feature maps, whose core idea is to compare the activation differences between foreground and background on the ID and OOD feature maps”, It is a little bit confused why Feature Sim compares the activation differences between foreground and background on the ID and OOD feature. More explanations should be provided.
>
> **A1:** Thanks for this comment. We apologize for not providing a clear description. In fact, for a sample, Feature Sim calculates the OOD score by examining the difference in activations between its foreground and background. We explain this operation through a toy example.
> We assume a $3 \times 3$ feature map of an ID sample as follows, where the middle pixel represents the foreground and the other pixels represent the background.
>
> | 0.1 | 0.1 | 0.1 |
> |-----|-----|-----|
> | 0.1 | 1.0 | 0.1 |
> | 0.1 | 0.1 | 0.1 |
>
> The mean of the feature map $\mu_{ID}=0.2$, then its Feature Sim Score is defined as $S_{ID}=(8\times|0.1-\mu_{ID}|+|1.0-\mu_{ID}|) / 9=0.18$
>
> For an OOD sample, based on our observations in the text, it should have lower foreground activation than an ID sample. Suppose its foreground is located at the bottom right corner, the feature map is represented as:
>
> | 0.1 | 0.1 | 0.1 |
> |-----|-----|-----|
> | 0.1 | 0.1 | 0.1 |
> | 0.1 | 0.1 | 0.4 |
>
> We get $\mu_{OOD}=0.13$, then its Feature Sim Score is defined as $S_{OOD}=(8\times|0.1-\mu_{OOD}|+|0.4-\mu_{OOD}|)/9=0.06$. Therefore, we have $S_{ID}=0.18 > S_{OOD}=0.06$.
>
>
> >**W2:**"Eq. (2) is equivalent to the GAP operator". Actually, Eq. 2 is not a GAP operator, GAP performs global pooling in the feature map, whereas Eq. 2 performs pooling across different channels.
>
> **A2:** Sorry for the misunderstanding. We want to show that Eq. 2 is equivalent to the GAP operator on on the spatial (height, width) axis. For each channel, a feature map with shape of (H, W) will be pooled as a single value. To be specific, the shape of f(x) is (B,C,H,W) and the shape of GAP(f(x)) is (B,C). We will add some details about the GAP operator in the updated version.
>
>
> >**W3:** In section A.1, the authors train the ResNet50 from scratch, while other comparison methods (Energy, ReAct, DICE ect.) use the off-the-shelf ResNet model for experiments. Therefore, the comparisons might be unfair, I recommend the author to utilize the off-the-shelf model.
>
> **A3:** Sorry for the **misunderstanding**. **In our experiments, we actually used the official pre-trained checkpoints provided by mmclassification**, which aligns with the performance of other comparison methods (Energy, ReAct, DICE, etc.). This ensures that our comparison with these methods is fair and consistent. The detailed description of the training process for the official pre-trained weights was included for clarity, but it seems to have caused some confusion.
>
>
> The reason we use the checkpoint provided by mmclassification rather than the official PyTorch one is that the official PyTorch does not provide checkpoints for resnet18 on cifar10 and cifar100. In order to ensure that all models are trained from the same framework and to avoid experimental unfairness due to differences in training methods, we use the official mmclassification checkpoint. Additionally, we have verified that its performance on the ImageNet-1k benchmark can align with the official PyTorch checkpoint.

---

> > ### Author Response · Authors · 2023-11-20
> >
> > >**W4:** The Threshold Activation module is quite similar to ReAct and ASH, which makes the novelty of the introduced TA module limited. Besides, the TA module decreases the ID classification accuracy significantly, even it can be remedied by using another classification branch.
> >
> > **A4:** Thanks for your suggestions! We already have a short comparison about these question in Section 2 final part and detailed explanations in Appendix E. For greater clarity and to aid the reader's understanding, we will provide detailed explanations:
> >
> > **Difference between our method and ReAct:**
> >
> > React believes that *abnormally high activations* in the fully connected layer can lead to classifier overconfidence, so it truncates all activations that are **higher than** the threshold. On the other hand, our Threshold Activation (TA) method starts from the phenomenon of activations in feature maps and truncates all activations in intermediate layers that are **lower than** the threshold while weakening the other activations. This approach weakens a large number of OOD features that could potentially generate abnormal activations, preventing the foreground parts of OOD images from producing excessively high activations. While React and TA seem to use a threshold to handle feature activations, their operational methods are entirely different.
> >
> > The Threshold Activation (TA) and ReAct methods handle features at different positions in the network, leading to opposite choices in retaining features. TA has sufficient network parameters following it to reprocess and restore features, allowing it to choose to retain **salient (high)** activations. On the other hand, ReAct has only one fully connected layer after it, limiting the network's ability to further process features. As a result, ReAct chooses to retain **stable (low)** activations. The differences in the types of features they handle (spatial features vs. classification features) and their positions in the network (intermediate convolutional layers vs. classifier layer) result in the adoption of completely different feature selection approaches between TA and ReAct.
> >
> > In addition, as shown in Table 2, using TA and ReAct together achieves better results. This indicates that TA and ReAct are complementary methods. TA's processing of convolutional features and ReAct's handling of fully connected classification features bring different gains to OOD detection. Experimental results demonstrate that these two gains can be combined, showing that the benefits of using TA and ReAct are cumulative.
> >
> > Based on the above content, we believe that Threshold Activation and ReAct have fundamental differences and TA should not be considered as incremental work of ReAct.
> >
> >
> > **Difference between our method and ASH:**
> >
> > ASH is an plug-in algorithm that requires existing score functions to work in conjunction with it, such as MSP and Energy.  Furthermore, while ASH claims to operate on feature map activations, it actually works well when placed in the penultimate layer of the classifier, so it does not effectively process feature map activations as claimed.
> >
> > Essentially, we believe that ASH truncates features, whereas our proposed TA weakens them.
> >
> > We have noticed that the core idea behind the three forms of ASH is to set the activations below the threshold to zero and preserve or even amplify those above the threshold. However, due to the characteristics of convolutional kernels, the remaining high OOD responses can "diffuse" back to other regions during the forward propagation of the network, recontaminating the "purified" feature maps (see Fig. 10 and Table 16 in the Appendix). This also explains why placing the ASH method in non-classifier positions leads to poor results.
> >
> > In contrast, TA has some similarities in design with ASH but carries a different essence. Firstly, we observe that the absolute strength of responses for ID features is higher than that of OOD features in the feature maps. Therefore, if we weaken both ID and OOD features by the same value 'k', the information preserved in ID features will be much more than that in OOD features (see Fig. 10 and Table 16 in the Appendix). Secondly, this weakening effect avoids the "contamination" issue seen in ASH's feature maps, ensuring that OOD, after being weakened, will not generate high activations to mislead classifier's outputs.
> >
> > **ID Accuracy Issue:**
> > **There is a misunderstanding about the ID performace.** It is important to note that our method **does not affect** the classification performance and only incurs a computational overhead of 7%, as evidenced in Table 8 of Appendix B.3. Specifically, we utilize the original features without the TA module for the ID classification task and obtain a copy of the mid-layer features at the TA insertion point for OOD detection. This ensures that the performance of the original classification task remains unaffected.

---

> > > ### Author Response · Authors · 2023-11-20
> > >
> > > >**Q1:** It is not clear why employ the absolute difference between the feature map and its mean value as OOD score. Have the authors considered to use the mean value of Gram Matrix as the similarity score, which I think should be better than the absolute difference. The Gram Matrix captures the similarity between different feature maps [1].
> > > [1] Image Style Transfer Using Convolutional Neural Networks.
> > >
> > >
> > > **A5:** Thanks for your comments! The use of mean absolute deviation as an OOD score is based on measuring the self-similarity of features, specifically the differences among each pixel within the feature. This approach stems from the need to quantify how distinct the foreground is from the background in an image. Since there's no absolute standard for each image's activations, only a relative one, the focus shifts to measuring the disparity between the image's own foreground and background. Various scores, including the most intuitive deviation (Standard deviation), are evaluated through ablation studies in Appendix B.10. It is found that mean absolute deviation offers the most balanced performance, while the simplest form of deviation (Standard deviation) also yields satisfactory results.
> > >
> > >
> > >
> > >
> > > |Statistics                   | iNaturalist |        |   SUN  |        | Places |        | Textures |        |         Average    |        |
> > > |:-----------------:|:-----------:|:------:|:------:|:------:|:------:|:------:|:--------:|:------:|:------------------:|:------:|
> > > |                   |    AUROC    |  FPR95 |  AUROC |  FPR95 |  AUROC |  FPR95 |   AUROC  |  FPR95 |        AUROC       |  FPR95 |
> > > | Mean absolute deviation   (baseline) |    96.86%   | 16.03% | 93.58% | 29.26% | 88.68% | 43.40% |  95.54%  | 20.48% |       93.67%       | 27.29% |
> > > |       Median absolute deviation      |    94.90%   | 24.51% | 93.75% | 28.52% | 90.71% | 38.38% |  87.17%  | 52.78% |       91.63%       | 36.05% |
> > > |        Standard deviation         |    96.70%   | 16.77% | 93.10% | 30.38% | 87.81% | 45.14% |  95.67%  | 19.89% |       93.32%       | 28.05% |
> > > |   Channel   mean absolute deviation   |    95.29%   | 25.35% | 94.72% | 25.77% | 90.52% | 38.50% |  86.01%  | 45.82% |       91.63%       | 33.86% |
> > >
> > >  *Channel  mean absolute deviation: using the mean value in channel dimension instead of spatial dimension to calculate mean absolute deviation.
> > >
> > > Following the suggestion in this comment, we design Feature Sim scores w.r.t. the mean value, standard deviation and mean absolute deviation of the gram matrix. The experimental results (Feature Sim with gram matrix + Threshold Activation + Energy) are shown below. Note that the gram matrix is calculated by matrix multiplication of flattend and normalized C5 features $(N,HW,C) \cdot(N,C,HW)->(N,HW,HW)$.
> > >
> > > |                          | iNaturalist |        |   SUN  |        | Places |        | Textures |        | Average |        |
> > > |:------------------------:|:-----------:|:------:|:------:|:------:|:------:|:------:|:--------:|:------:|:-------:|:------:|
> > > |                          |    AUROC    |  FPR95 |  AUROC |  FPR95 |  AUROC |  FPR95 |   AUROC  |  FPR95 |  AUROC  |  FPR95 |
> > > |           Mean           |    25.67%   | 99.91% | 55.15% | 98.12% | 58.55% | 97.86% |  20.93%  | 99.77% |  40.07% | 98.91% |
> > > |    Standard deviation    |    96.23%   | 19.46% | 88.91% | 50.77% | 86.30% | 60.63% |  96.22%  | 14.59% |  91.92% | 36.36% |
> > > | Mean absolute deviation  |    95.79%   | 22.07% | 88.01% | 54.59% | 85.47% | 63.54% |  95.55%  | 17.73% |  91.21% | 39.48% |
> > >
> > > It is interesting that the mean value of gram matrix does not work at all, while the standard deviation and mean absolute deviation of gram matrix work just fine. We believe this is because when the difference in foreground and background activation is significant (ID sample), the gram matrix will also show a large deviation, and vice versa. This phenomenon also confirms our foreground-background hypothesis.
> > >
> > > For the mean of the gram matrix, since the cosine similarity between features changes between $[-1,1]$, the meaning of feature activation strength has been lost. Therefore, directly measuring the mean of the gram matrix may have no physical meaning. We will add relevant experiments and discussions to the appendix of the paper.

---

> > > > ### Comment · Reviewer_sGRf · 2023-11-22
> > > > **Response to Authors Rebuttal**
> > > >
> > > > I appreciate the authors' addressing of my concerns and I will raise my rating.
> > > > However, I still feel that the proposed method is similar to the previous works such as ASH.  The novelty of this paper is limited due to the just subtle differences in thresholds

---

> > > > > ### Author Response · Authors · 2023-11-22
> > > > > **Thanks for raising your score!**
> > > > >
> > > > > Thank you very much for your valuable suggestions, which have helped improve our paper. Regarding the issue of innovation, we wish to restate the following points:
> > > > >
> > > > > Firstly, Threshold Activation differs in motivation from ASH, as it leverages the varying abilities of deep networks to process and recover ID and OOD features, thereby enhancing the distinction between ID and OOD.
> > > > >
> > > > > Secondly, our most significant innovation lies in using the activation differences between foreground and background in feature maps to identify OOD samples. Threshold Activation is merely a module to amplify this difference. We hope the reviewers will prioritize our main contribution, Feature Sim.
> > > > >
> > > > > We sincerely hope that the reviewers will recognize our work's contribution to discovering and detecting OOD in Feature Maps in the upcoming phases. Thank you again for your review.

---

### Official Review · Reviewer_gWG8 · 2023-11-01

**Soundness:** 2 fair
**Presentation:** 2 fair
**Contribution:** 2 fair
**Rating:** 5
**Confidence:** 4

**Summary:**

The paper proposes a post-hoc method for out-of-distribution detection. The authors observe that ID data is more robust than OOD data and the gap between foreground and background features is more prominent. Based on these observations, the authors propose a new OOD score function based on the spatial feature maps, and a threshold activation technique to further boost the performance. Experiments are conducted on various OOD detection benchmarks and some results are appealing.

**Strengths:**

1. The proposed method is simple and easy to implement.
2. The authors conduct comprehensive experiments to evaluate the method.
3. The method achieves state-of-the-art results when combined with other techniques.

**Weaknesses:**

1. he method claims to leverage spatial clues in the feature maps. However, no spatial information is actually used in the final OOD score calculation (cf. Eq. (3)).

2. The method is mainly based on the observation that the activation difference between foreground and background is more prominent for ID data. However, the generalizability of this observation remains unverified. Notably, most of the visualizations of OOD data in Figure 6 display small foreground objects, which calls into question the applicability of this method to a broader range of OOD scenarios.

3. The theoretical analysis of the method assumes that the features of foreground and background follow two Gaussian distributions. This is a strong assumption that may not hold in realistic settings.

4. As indicated in Table 1, this method requires integration with other OOD detection techniques to achieve state-of-the-art results. This raises two concerns: Firstly, combining different OOD functions could understandably lead to a performance boost, making it unclear how much of the improvement is attributable to the proposed method itself. Secondly, the aggregation parameters appear to require meticulous tuning, as shown in Appendix A.1, which casts doubt on the method's ease of use in real-world applications.

5. The paper claims that the proposed method is adaptable to other tasks such as semantic segmentation and object detection. However, the experimental setup is puzzling. The method appears to focus on distinguishing between image-level ID and OOD, which deviates from these standard tasks, such as OOD segmentation [1]. Besides, it remains unclear the unique advantages of this method in these application areas.  For example, one could easily use a simple average of existing pixel-wise OOD scores as the image-level OOD score for the segmentation task.  There are no comparisons with any baseline methods in Table 6.

[1] Chan Robin, et al. Segmentmeifyoucan: A benchmark for anomaly segmentation. NeurIPS, 2021.

**Questions:**

See detailed comments/concerns above.

---

> ### Author Response · Authors · 2023-11-20
>
> >**W1:** The method claims to leverage spatial clues in the feature maps. However, no spatial information is actually used in the final OOD score calculation (cf. Eq. (3)).
>
> **A1:** Thanks for your suggestions! Our approach utilizes the spatial proportions and relative activation strengths of the foreground and background. However, upon further consideration, especially after a reviewer's reminder, we acknowledge that our method does not utilize a strong spatial positional relationship. As a result, employing the term 'spatial clues' might cause confusion for readers, and we intend to revise this phrasing.
>
> >**W2:** The method is mainly based on the observation that the activation difference between foreground and background is more prominent for ID data. However, the generalizability of this observation remains unverified. Notably, most of the visualizations of OOD data in Figure 6 display small foreground objects, which calls into question the applicability of this method to a broader range of OOD scenarios.
>
>
> **A2:** Thanks for pointing out this issue. In fact, the oracle method should segment the foreground and background areas of the image, and then separately calculate the average activation intensity within these areas, thereby obtaining the difference in activation between the foreground and background, making the method unaffected by the scale of the object. However, we cannot obtain segmentation annotations for an image in advance.
> Fortunately, our method is not overly sensitive to object scale, because the most of the OOD foreground activations are **quite lower** than that of ID images. Therefore, Feature Sim is robust to changes in scale.
>
> To verify this phenomenon, we plan to locally magnify images where the foreground occupies a smaller proportion, observing the impact of size changes on the OOD Score of these samples. The results will be updated here once we complete the experiments！
>
> >**W3:** The theoretical analysis of the method assumes that the features of foreground and background follow two Gaussian distributions. This is a strong assumption that may not hold in realistic settings.
>
> **A3:** Thanks for your comments! In the field of OOD detection, it’s common to use Gaussian distributions to model features and aid in explaining the algorithm’s mechanism [1][2]. The Gaussian Mixture Model (GMM) we employ is more general and can cover more types of features compared to Gaussian distributions.
>
> Regarding the GMM modeling of foreground-background, we reiterate that the one-dimensional GMM only represents the probability densities of activation components for feature foreground and background, without encoding their spatial relationships. Therefore, we do not consider GMM to be a stringent assumption. Moreover, in Appendix B.10, we employ the foreground-background differences fitted by GMM directly as the OOD score. Experimental results on the ImageNet-1k benchmark are 90.83% AUROC and 35.08% FPR95, which effectively substantiates the rationale behind our GMM-based modeling.
>
> Our intention with employing a toy model is to facilitate reader comprehension of how our module operates. A more complex model that closely mimics real-world scenarios does not yield an analytical solution but only results from numerical simulations. The GMM we utilized effectively describes the histogram distributions of the foreground and background in most scenarios, as illustrated by the blue and orange lines in Figures 7 and 8 of the Appendix. As indicated in Section 7, titled "Understanding Feature Sim from the Perspective of GMM," our objective isn't to provide a rigorous theoretical proof in this section, but rather to enhance the reader's understanding of the Feature Sim (FS) working mechanism.

---

> > ### Author Response · Authors · 2023-11-20
> >
> > >**W4:** As indicated in Table 1, this method requires integration with other OOD detection techniques to achieve state-of-the-art results. This raises two concerns: Firstly, combining different OOD functions could understandably lead to a performance boost, making it unclear how much of the improvement is attributable to the proposed method itself. Secondly, the aggregation parameters appear to require meticulous tuning, as shown in Appendix A.1, which casts doubt on the method's ease of use in real-world applications.
> >
> > **A4:** Thanks for your suggestions!
> >
> > **Method combination:**
> > Many previous methods [1,3,4] have also relied on combinations to achieve improved results . For instance, recent approaches like Ash [3], ReAct [1], and DICE [4] are not indenpendent OOD detectors. They enhance OOD performance by modifying network hidden states and then influencing OOD scores like Energy [5]. In contrast, our proposed Feature Sim (FS) is an OOD score function. The integration of Threshold Activation (TA) further enhances the differentiation between ID and OOD samples, thereby improving the FS scores as well as those of previous OOD detectors (e.g., Energy, MSP, etc.) as demonstrated in our experiments (see the last three rows of Table 1). We consider the plug-and-play characteristic of FS and TA to be strengths rather than weaknesses. Moreover, comparing FS+TA+Energy with ASH in Table 2, our method FS+TA+Energy can still achieve the state-of-the-art, showing the high performance improvement provided by our method.
> >
> > **Hyperparameters:**
> > We would like to clarify that the selection of the composite $\lambda$ value is not sensitive  because our proposed TA+FS method and the target method have different dynamic ranges. For instance, ODIN varies between [0.001001, 0.001029] (within a $3\sigma$ range, similarly for the others), while Energy varies between [4.04, 25.46]. When combining our method with the two methods respectively, $\lambda$ can differ by hundreds or even thousands of times due to the disparate dynamic ranges.
> >
> > The introduction of $\lambda$ is, in fact, the most intuitive approach as it normalizes the variation ranges of the two methods to the same interval, making the fusion weights approximately equal. It is important to note that this normalization is independent of the specific data distribution and only depends on the characteristics of the two methods involved in the fusion. Hence, there is no issue of test set leakage.
> >
> > Additionally, $\lambda$ is not a meticulously searched parameter (see Fig.4 in the Appendix); we simply adjust the dynamic range of the two scores to be of the same order of magnitude, allowing the fused score function to adequately incorporate opinions from both convolutional features and the classifier, rather than having one dominate absolutely. In fact, $\lambda$ is robust to variations over a wide range.

---

> > > ### Author Response · Authors · 2023-11-20
> > >
> > > >**W5:** The paper claims that the proposed method is adaptable to other tasks such as semantic segmentation and object detection. However, the experimental setup is puzzling. The method appears to focus on distinguishing between image-level ID and OOD, which deviates from these standard tasks, such as OOD segmentation [1]. Besides, it remains unclear the unique advantages of this method in these application areas. For example, one could easily use a simple average of existing pixel-wise OOD scores as the image-level OOD score for the segmentation task. There are no comparisons with any baseline methods in Table 6.
> > > [1] Chan Robin, et al. Segmentmeifyoucan: A benchmark for anomaly segmentation. NeurIPS, 2021.
> > >
> > > **A5:** Thank you for your comments! We apologize for not clearly detailing the experimental setup in the main text. Specifically, our approach involves conducting OOD detection directly before object detection and segmentation tasks, rejecting obvious OOD samples upfront. Firstly, we believe preemptively discarding clear OOD samples before focusing on granular tasks like detection and segmentation is beneficial. This is because task-specific OOD detection (at object or pixel level) is more costly and requires custom designs for each visual task. For example, if a network's task is to detect apples in orchard images and a camera consistently captures the ground due to being loose, identifying this OOD image before performing specific detection tasks can significantly reduce risks and costs associated with OOD samples.
> > >
> > > Secondly, the current domain of OOD detection lacks a unified Image-Level OOD benchmark and baselines tailored for various visual tasks. However, our method demonstrates the potential to consistently detect OOD samples across different tasks by assessing whether a network's feature extractor is "familiar" with an input sample. While we have only tested on Object Detection and Semantic Segmentation tasks, we believe our approach can generalize to a wider range of tasks, such as depth estimation, pose recognition, and pedestrian re-identification. To our knowledge, there are no specialized OOD detection methods for these tasks yet, which is why we describe FS+TA as a unified method.
> > >
> > > We are conducting experiments about using a simple average of existing pixel-wise OOD scores as the image-level OOD score for the segmentation task, and the results will be updated here. By the way, we think that this baseline proposed by the reviewer is only applicable when the network has a semantic segmentation head. In the field of object detection, this would require a separate design.
> > >
> > > **Reference**
> > >
> > > [1] Sun, Yiyou, Chuan Guo, and Yixuan Li. “React: Out-of-distribution detection with rectified activations.” Advances in Neural Information Processing Systems 34 (2021): 144-157.
> > >
> > > [2] Ming, Yifei, Hang Yin, and Yixuan Li. “On the impact of spurious correlation for out-of-distribution detection.” Proceedings of the AAAI Conference on Artificial Intelligence. Vol. 36. No. 9. 2022.
> > >
> > > [3] A. Djurisic, N. Bozanic, A. Ashok, and R. Liu. Extremely simple activation shaping for out-of- distribution detection. ICLR 2023.
> > >
> > > [4] Sun, Yiyou, and Yixuan Li. "Dice: Leveraging sparsification for out-of-distribution detection." European Conference on Computer Vision. Cham: Springer Nature Switzerland, 2022.
> > >
> > > [5] Weitang Liu, Xiaoyun Wang, John D. Owens, and Yixuan Li. Energy-based out-of-distribution detection. In NeurIPS, 2020.

---

> > > > ### Author Response · Authors · 2023-11-22
> > > > **Updated experiments for the baseline in the segmentation task**
> > > >
> > > > We use a simple average of existing pixel-wise OOD scores as the image-level OOD score for the segmentation task and the results are shown below. We will also update the baseline in our revised paper.
> > > >
> > > >
> > > >
> > > >
> > > > |  Method  | iNaturalist |            |     SUN    |            |   Places   |            |  Textures  |            |   Average  |            |
> > > > |:--------:|:-----------:|:----------:|:----------:|:----------:|:----------:|:----------:|:----------:|:----------:|:----------:|:----------:|
> > > > |          |    AUROC    |    FPR95   |    AUROC   |    FPR95   |    AUROC   |    FPR95   |    AUROC   |    FPR95   |    AUROC   |    FPR95   |
> > > > |    MSP   |      82.12  |     66.25  |     62.90  |     89.25  |     64.44  |     91.00  |     83.80  |     50.75  |     73.32  |     74.31  |
> > > > | NegLabel |  **95.44** | **25.25** | **85.31** | **63.00** | **78.61** | **72.25** | **95.08** | **22.50** | **88.61** | **45.75** |
> > > >
> > > > The core pseudo codes of the implementation are as follows:
> > > >
> > > > outputs = self.segmentor.inference(input["img"], input["img_metas"])  # outputs: (B, C, H, W)
> > > >
> > > > outputs = outputs.max(dim=1)[0]  # MSP in category dimension. (B, H, W)
> > > >
> > > > confs = outputs.mean(dim=(-1, -2))  # Average among all pixels (B,)

---

> ### Author Response · Authors · 2023-11-22
> **Updated experiments for the object scale**
>
> Reviewer gWG8 is concerned that the foreground sizes in the OOD datasets we selected are small, resulting in lower OOD scores. Reviewer gWG8 is curious whether the OOD score would increase if the foreground size in the OOD image increases, thus affecting the OOD detection performance of FS.
>
> Among the four OOD datasets, only iNaturalist is object-centric, so it is most susceptible to the impact of object scale. Therefore, we use iNaturalist as an example for our experiments. We zoom in the OOD image w.r.t. the image center (w/2, h/2) and then resize it to 224. This way, we will get foregrounds of different sizes. We perform continuous zoom-in operations on the iNaturalist dataset, without altering the ID dataset. The experimental results of FS+TA are shown below.
>
> | Scale Ratio | iNaturalist |        |
> |-------------|:-----------:|:------:|
> |             |    AUROC    |  FPR95 |
> |           1 |      96.86% | 16.03% |
> |         1.1 |      97.84% | 10.90% |
> |         1.2 |      97.92% | 10.57% |
> |         1.3 |      98.08% |  9.78% |
> |         1.4 |      98.17% |  9.18% |
> |         1.5 |      98.25% |  8.62% |
> |         1.6 |      98.30% |  8.25% |
> |         1.7 |      98.33% |  8.18% |
> |         1.8 |      98.34% |  8.33% |
> |         1.9 |      98.34% |  8.31% |
> |           2 |      98.48% |  7.66% |
>
> The experimental results show that when the foreground size of OOD images increases, the performance of OOD detection improves. Our detailed analysis of this phenomenon is as follows:
>
> * During the continuous enlargement of the foreground size, a mismatch occurs between the size of the object and the receptive field of the network, leading to a further reduction in the network's response to the foreground. This causes the OOD scores of iNaturalist to decrease, resulting in an increase in the OOD detection performance.
> * The increase in the proportion of the foreground in OOD images does not enhance the activation intensity of the foreground. The experimental phenomenon is consistent with our previous explanation.
>
> Moreover, we also examine some samples shown in Figure 7, and their OOD scores also decrease as the size of the foreground increased. We will update these discussions in the revised paper.

---

> > ### Comment · Reviewer_gWG8 · 2023-11-23
> > **Thanks for your response**
> >
> > Thanks for the detailed response and additional results.
> > - W1-3: As the reply indicates, it seems that the paper requires some significant revision.
> > - W4: The dynamic range in Fig 4 is pretty narrow. The analysis does not fully convince me.
> > - W5: Thanks for the clarification. The added results partially addressed this concern. However, it is uncommon/infeasible to discard input images in many real-world semantic segmentation and object detection tasks. It is a bit overclaimed.

---

> > > ### Author Response · Authors · 2023-11-23
> > >
> > > > W1-3: As the reply indicates, it seems that the paper requires some significant revision.
> > >
> > > **A1:** Thank you very much for your valuable feedback. We have made revisions to our paper and added some discussions in our latest uploaded PDF document.
> > >
> > > >W4: The dynamic range in Fig 4 is pretty narrow. The analysis does not fully convince me.
> > >
> > > **A2:** Thanks for the comment. We would like to point out that in Figure 4, the value of $\lambda$ in subfigure (a) has changed from 0.006 to 0.03, a fivefold increase. As the Energy scores are quite large,  the dynamic range of  $\lambda$ appear relatively small. The performance of Energy with $\lambda$ within this fivefold range remains stable, indicating that $\lambda$ is a robust hyperparameter.
> > >
> > >
> > > >W5: Thanks for the clarification. The added results partially addressed this concern. However, it is uncommon/infeasible to discard input images in many real-world semantic segmentation and object detection tasks. It is a bit overclaimed.
> > >
> > > **A3:** Thanks for the comment. Despite the differing viewpoints between the reviewer and us, we maintain our opinion that OOD detection for general tasks is meaningful. We would like to reiterate the motivation of  this setting for universal OOD detection: first, versatility, that is, plug-and-play. We believe that no matter what task the model is designed for, as long as we can obtain the parameters of the backbone, we can access our method for OOD detection and filtering. The second is security: by preprocessing images in a global view before engaging in specific tasks, we can effectively mitigate the risks associated with unknown inputs. This setting enhances the safety of the model before it delves into the particulars of a given task.

---

### Author Response · Authors · 2023-11-22
**Summary of the changes in the updated paper**

We sincerely thank the reviewers for their insightful suggestions. In response, we have revised our manuscript, highlighting the changes in blue. The key modifications made in the updated version of our paper are outlined below:

* We revise the expression about "spatial clues". Thanks for the suggestion of Reviewer gWG8.
* We add a discussion about the object scale in Appendix B.13. Thanks for the suggestion of Reviewer gWG8.
* We add a baseline for the segmentationtask in Appendix B.7. Thanks for the suggestion of Reviewer gWG8.
* We add add some details about the GAP operator in Section 5.1. Thanks for the suggestion of Reviewer sGRf and NfBj.
* We add the discussion about the Gram Matrix in Appendix B.10. Thanks for the suggestion of Reviewer sGRf.
* We add the near OOD experiments in Appendix B.9. Thanks for the suggestion of Reviewer NfBj.
* We add some details about the position of FS and TA in Appendix A.1. Thanks for the suggestion of Reviewer NfBj.

We kindly ask the reviewers to review our responses and continue the discussion. Feel free to ask more questions or adjust your rating. We are looking forward to your reply.

---

### Meta-Review · Area_Chair_DBhx · 2023-12-11

**Metareview:**

This work received four borderline scores. Though there are some merits in this work, reviewers still have a series of concerns about this work. They pointed out that the work uses some strong assumptions which may not hold in practical applications. Also, it is infeasible to discard the input images in real-world segmentation and detection tasks. Besides, the proposed approach is similar to previous works such as ASH, and thus the novelty and contributions of this paper seem to be limited. The proposed FS+TA also performs worse than the current state-of-the-art ASH. Considering the weaknesses in terms of novelty, performance, used assumptions, the area chair recommends rejection for this paper. Authors are encouraged to improve the paper following reviewers' comments and submit it to another top-tier venue.

**Justification For Why Not Higher Score:**

The work uses some strong assumptions which may not hold in practical applications. Also, it is infeasible to discard the input images in real-world segmentation and detection tasks. Besides, the proposed approach is similar to previous works such as ASH, and thus the novelty and contributions of this paper seem to be limited. The proposed FS+TA also performs worse than the current state-of-the-art ASH.

**Justification For Why Not Lower Score:**

N/A

---

### Decision · Program_Chairs · 2024-01-16

Reject